# A Certified Unlearning Approach without Access to Source Data

**Umit Yigit Basaran** [1]   **Sk Miraj Ahmed** [2]   **Amit Roy-Chowdhury** [1]   **Başak Güler** [1]

## Abstract

With the growing adoption of data privacy regulations, the ability to erase private or copyrighted information from trained models has become a crucial requirement. Traditional unlearning methods often assume access to the complete training dataset, which is unrealistic in scenarios where the source data is no longer available. To address this challenge, we propose a certified unlearning framework that enables effective data removal without access to the original training data samples. Our approach utilizes a surrogate dataset that approximates the statistical properties of the source data, allowing for controlled noise scaling based on the statistical distance between the two. While our theoretical guarantees assume knowledge of the exact statistical distance, practical implementations typically approximate this distance, resulting in potentially weaker but still meaningful privacy guarantees. This ensures strong guarantees on the model's behavior post-unlearning while maintaining its overall utility. We establish theoretical bounds, introduce practical noise calibration techniques, and validate our method through extensive experiments on both synthetic and real-world datasets. The results demonstrate the effectiveness and reliability of our approach in privacy-sensitive settings.

## 1. Introduction

Machine learning models have achieved remarkable success across a wide range of applications by leveraging large-scale datasets. However, growing concerns about data privacy and regulatory requirements—such as the General Data Protec-

tion Regulation (GDPR, 2016), California Consumer Privacy Act (CCPA, 2018), or the Canadian Consumer Privacy Protection Act (CPPA, 2023)—have led to a pressing demand for mechanisms that allow the removal of specific data points from trained models.

Among these mechanisms, certified unlearning has emerged as a cornerstone, formalizing data removal with rigorous guarantees. In contrast to heuristic methods, certified unlearning ensures that the influence of removed data is provably eliminated from the model, offering both privacy compliance and practical utility. This is typically achieved by bounding the statistical discrepancy between a model retrained without the forget data and an approximated model that simulates this retraining. While fully retraining a model from scratch on the retained data is a straightforward and rigorous solution, it becomes computationally prohibitive for large-scale models and frequent deletion requests. Certified unlearning instead provides a practical alternative: it enables efficient, provably correct data removal without exhaustive retraining (Guo et al., 2019; Neel et al., 2021; Sekhari et al., 2021; Chien et al., 2024; Zhang et al., 2024).

Existing certified unlearning methods generally focus on two key aspects. First, they approximate the retrained model using an efficient technique, often leveraging a single-step Newton update (Guo et al., 2019; Sekhari et al., 2021; Zhang et al., 2024). Second, they inject noise based on differential privacy (Dwork, 2006) principles—commonly via a Gaussian mechanism—to ensure that the retrained and unlearned models are statistically indistinguishable. We adopt this single-step update approach in our work as it strikes a balance between efficiency, theoretical grounding, and empirical effectiveness, as supported by prior studies. Despite their promise, these methods often rely on the assumption that the source data remains accessible during unlearning.

A critical challenge arises when the unlearning mechanism has no access to the source data samples. This limitation may stem from privacy constraints, as the original model may have been trained by a different organization or a third party; resource restrictions, as old data may be deleted due to memory constraints; or regulatory barriers, as data retention policies or security concerns may prohibit storing the source data. As a result, existing unlearning methods that rely on access to the source data become impractical or infeasible.

---

[1]Department of Electrical and Computer Engineering, University of California, Riverside, CA, USA [2]Brookhaven National Laboratory, Upton, NY, USA. Correspondence to: Umit Yigit Basaran <umit.basaran@email.ucr.edu>, Sk Miraj Ahmed <sahme047@ucr.edu>, Amit Roy-Chowdhury <amitrc@ucr.edu>, Başak Güler <bguler@ece.ucr.edu>.

*Proceedings of the 42^{nd} International Conference on Machine Learning*, Vancouver, Canada. PMLR 267, 2025. Copyright 2025 by the author(s).

Consequently, a key open question emerges:

- *What if the unlearning mechanism has no access to the original training samples, and instead must rely entirely on a surrogate dataset that mimics these statistical properties to achieve certified unlearning?*

This problem is related to zero-shot unlearning, where no information about the true dataset is available during forgetting, beyond the model. While several relevant methods (Foster et al., 2024; Chundawat et al., 2023; Cha et al., 2023) have been proposed, no theoretical guarantees exist.

A more tractable scenario in practice is when the unlearning mechanism has access to a *surrogate dataset* that mimics the statistical properties of the source data up to a specified level of fidelity. Summary statistics, learned from this surrogate dataset, are then used to guide the unlearning process. For instance, consider a case where a person requests their data to be deleted, but the organization no longer retains the original training data due to regulatory or resource limitations. Instead, the organization uses publicly available data or previously generated surrogate datasets that closely resemble the source data distribution to facilitate the unlearning process. Alternatively, the individual may provide a new set of data samples (e.g., images)—distinct from the source data but with some statistical discrepancy—specifically to facilitate the unlearning process.

In this work, we address precisely this scenario. *We study certified unlearning when the unlearning mechanism has no access to source (retain) samples, but instead relies on samples from a surrogate dataset that mimics the source data up to a fidelity criterion.* Our goal is to establish formal certified unlearning guarantees during unlearning, and how the unlearning performance changes as a function of the distance between the surrogate and source data.

**Main Result.** We propose a certified unlearning framework that does not require access to the original retain data samples. Instead, we leverage a surrogate dataset where the samples are generated from a distribution that may be different from the original. By carefully scaling the noise based on the statistical distance between the source and surrogate datasets, we provide rigorous indistinguishability guarantees on how closely the unlearned model mimics the truly retrained model. These guarantees are explicitly dependent on the distance between the source and surrogate data.

Formally, let $\mathcal{D}$ denote the source dataset drawn from a distribution $\rho$, whereas $\mathcal{D}_s$ denotes a surrogate dataset drawn from a distribution $\nu$. Both distributions share the support $\mathcal{X} \times \mathcal{Y}$, where $\mathcal{X}$ represents the feature space and $\mathcal{Y}$ represents the label space. Suppose we want to remove a set of samples $\mathcal{D}_u$ from $\mathcal{D}$. We define $\boldsymbol{w}_r^*$ as the model retrained from scratch on the original retain data $\mathcal{D}_r = \mathcal{D} \backslash \mathcal{D}_u$ and $\widehat{\boldsymbol{w}}_r$ as the model produced by our unlearning mechanism

that uses $\mathcal{D}_s$ in place of the source dataset $\mathcal{D}$. Throughout the paper, we focus on classification models.

Under typical assumptions (Assumption 4.1) about the loss function used to train the model, we prove that the norm of the difference between the model retrained from scratch over the original retain data, $\boldsymbol{w}_r^*$, and the model approximated with our unlearning mechanism using the surrogate dataset, $\widehat{\boldsymbol{w}}_r$, is upper bounded as (Theorem 4.2),

$$\|\boldsymbol{w}_r^* - \widehat{\boldsymbol{w}}_r\|_2 \leq \Delta$$

where $\Delta$ is a function of the statistical distance between the two distributions $\rho$ and $\nu$. Then, by carefully scaling the noise as a function of the statistical distance between $\rho$ and $\nu$, we achieve certified unlearning without direct access to the retain data. Technical details are provided in Section 4.

**Contributions.** Our contributions are summarized below.

- We propose a certified unlearning mechanism to forget samples drawn from a given distribution without having access to the true retain samples (from the source distribution). Instead, it relies on a surrogate dataset $\mathcal{D}_s$ sampled from a different distribution to ensure certified unlearning. This is the first work to provide certified unlearning guarantees when the source data is not available, offering a practical solution in scenarios where direct access to the source data is restricted.
- We establish rigorous certified unlearning guarantees that hinge on the statistical distance between the source data distribution $\rho$ and the surrogate data distribution $\nu$. Our main theorem ensures that the influence of the unlearned data points is effectively removed while preserving a provable bound.
- For scenarios when the statistical (distance) information is not readily available between the source and surrogate datasets, we introduce a heuristic to approximate the distance between $\rho$ and $\nu$. Our approach uses only the model and surrogate dataset, without any information about the source statistics, making it well-suited for resource-limited environments.
- We provide an extensive set of experiments, on both synthetic and real-world datasets, to demonstrate the effectiveness of our approach. In particular, we show how our noise-calibration procedure ensures certified unlearning while maintaining utility comparable to methods that assume full access to source data.

## 2. Related Works

***Certified Unlearning.*** Certified unlearning has become a key mechanism for data removal, offering rigorous privacy guarantees while avoiding full retraining costs. Existing methods typically rely on single-step Newton updates, influence functions (Guo et al., 2019; Sekhari et al., 2021; Zhang

et al., 2024), or projected gradient descent algorithms (Neel et al., 2021; Chien et al., 2024) combined with a randomization mechanism for statistical indistinguishability.

***Source-Free Unlearning.*** Zero-shot machine unlearning (Chundawat et al., 2023) removes data influence using only model weights and the forget set, using noise-based error maximization and gated knowledge transfer. Another method (Cha et al., 2023) employs adversarial sample generation to preserve decision boundaries while applying gradient ascent on forget data. JiT unlearning (Foster et al., 2024) fine-tunes models with perturbed samples to reduce reliance on forget instances. Bonato et al. (Bonato et al., 2025) modify feature vectors to align forgotten data with the nearest incorrect class, using a surrogate dataset in source-free settings. Despite some recent work in zero-shot/source-free unlearning, formal certified guarantees remain an open problem. We address this by ensuring such guarantees using a surrogate dataset in the absence of source data.

## 3. Preliminaries and Problem Formulation

**Certified Unlearning.** Machine unlearning removes the influence of specific data points from a trained model while preserving its performance on the retained data. Certified unlearning formalizes this concept by offering rigorous probabilistic guarantees on the correctness and reliability of the unlearning process. In essence, it ensures that the adjusted model behaves as though it was fully retrained without the removed data, with bounded error relative to the fully retrained model. This contrasts with empirical unlearning techniques, which may be easier to implement but lack any formal assurances about the fidelity of the resulting model.

Let the original dataset $\mathcal{D}$ contain samples $\{\boldsymbol{x}, y\}_{i=1}^{n}$, each sampled from the joint distribution $\rho$ with the support set $\mathcal{X} \times \mathcal{Y}$. Let the set of samples to be unlearned, $\mathcal{D}_u \subset \mathcal{D}$, have size $|\mathcal{D}_u| = m$. Then, the retain set $\mathcal{D}_r = \mathcal{D} \setminus \mathcal{D}_u$ has size $|\mathcal{D}_r| = n - m$. A learning algorithm $\mathcal{A}$ takes $\mathcal{D}$ as input and outputs a model $\boldsymbol{w}^* = \mathcal{A}(\mathcal{D})$, which minimizes the expected loss $\mathbb{E}_{(\boldsymbol{x}, y) \sim \rho}\left[\mathcal{L}((\boldsymbol{x}, y), \boldsymbol{w})\right]$, where $\mathcal{L}((\boldsymbol{x}, y), \boldsymbol{w})$ measures the error of the model $\boldsymbol{w}$ on the data sample $(\boldsymbol{x}, y)$.

Having the trained model parameters $\boldsymbol{w}^*$, unlearning can be examined under exact and approximate approaches:

***1. Exact unlearning*** involves retraining the model from scratch on the retained data $\mathcal{D}_r$ (Bourtoule et al., 2021; Ullah et al., 2021; Dukler et al., 2023), which guarantees complete removal of $\mathcal{D}_u$'s influence but is often computationally infeasible. This motivated approximate unlearning methods to replicate exact unlearning at a reduced cost.

***2. Certified Approximate Unlearning*** provides a practical approach to certified unlearning by relaxing the requirement of retraining from scratch. Instead, it modifies the trained

model $\boldsymbol{w}^*$ directly so the influence of $\mathcal{D}_u$ is effectively removed. The goal is to construct an updated model $\boldsymbol{w}_r$ that closely approximates the retrained model $\boldsymbol{w}_r^*$, while ensuring strong statistical indistinguishability guarantees.

Most existing certified approximate unlearning methods employ techniques such as influence functions (Guo et al., 2019), second-order Newton updates (Sekhari et al., 2021; Zhang et al., 2024), or other optimization methods (Chien et al., 2024; Neel et al., 2021) to efficiently adjust $\boldsymbol{w}^*$. By incorporating carefully calibrated randomness, often through a Gaussian mechanism inspired by differential privacy (Dwork, 2006), approximate unlearning ensures that the statistical properties of $\boldsymbol{w}_r$ align with those of $\boldsymbol{w}_r^*$, up to a specified certification budget. In these approaches, an upper bound is placed on the norm of the difference between $\boldsymbol{w}_r^*$ and $\boldsymbol{w}_r$, which in turn determines the required noise variance. Details of the relation between differential privacy and certified unlearning are provided in Appendix B.

Formally, given a model $\boldsymbol{w}^*$ trained on dataset $\mathcal{D}$, the samples to be unlearned $\mathcal{D}_u$, and additional statistical information $\mathcal{S}(\mathcal{D})$ about $\mathcal{D}$, the unlearning mechanism $\mathcal{U}$ produces an updated model $\boldsymbol{w}_r$. The mechanism $\mathcal{U}$ satisfies $(\epsilon, \delta)$-certified unlearning if it adheres to the following guarantees.

**Definition 3.1** ($(\epsilon, \delta)$-Certified Unlearning (Sekhari et al., 2021))**.** Given a learning mechanism $\mathcal{A}$ defined over the hypothesis space $\mathcal{H}$, an unlearning mechanism $\mathcal{U}$ guarantees $(\epsilon, \delta)$ certified unlearning if and only if $\forall \mathcal{T} \subseteq \mathcal{H}$,

$$
\begin{aligned}
&\Pr\left(\mathcal{U}(\mathcal{D}_u, \mathcal{A}(\mathcal{D}), \mathcal{S}(\mathcal{D})) \in \mathcal{T}\right) \\
&\quad \leq e^\epsilon \Pr\left(\mathcal{U}(\emptyset, \mathcal{A}(\mathcal{D}_r), \mathcal{S}(\mathcal{D}_r)) \in \mathcal{T}\right) + \delta, \\
&\Pr\left(\mathcal{U}(\emptyset, \mathcal{A}(\mathcal{D}_r), \mathcal{S}(\mathcal{D}_r)) \in \mathcal{T}\right) \\
&\quad \leq e^\epsilon \Pr\left(\mathcal{U}(\mathcal{D}_u, \mathcal{A}(\mathcal{D}), \mathcal{S}(\mathcal{D})) \in \mathcal{T}\right) + \delta.
\end{aligned}
$$

where $\mathcal{S}$ is a mechanism that returns statistical information about the given dataset to guide unlearning $\mathcal{U}$.

Definition 3.1 ensures that the updated model $\boldsymbol{w}_r$ is statistically indistinguishable from the retrained model $\boldsymbol{w}_r^*$, up to the parameters $(\epsilon, \delta)$. We adopt the unlearning definition from (Sekhari et al., 2021) due to its versatility and practical benefits. Unlike earlier definitions (Ginart et al., 2019), which rely on the unlearning algorithm being inherently randomized even when no deletions occur.

**Second-order unlearning.** For certified unlearning, $\mathcal{S}(\cdot)$ typically represents detailed information about the true data samples (Guo et al., 2019; Neel et al., 2021; Sekhari et al., 2021; Chien et al., 2024; Zhang et al., 2024). A common approach is to utilize the second-order Newton update, for which $\mathcal{S}(\cdot)$ refers to the Hessian of $\mathcal{D}$, evaluated on the trained model $\boldsymbol{w}^*$ (Sekhari et al., 2021; Zhang et al., 2024). These approaches follow a general methodology consisting

of updating the model with a single-step Newton update,

$$\boldsymbol{w}_r \leftarrow \boldsymbol{w}^* - \frac{m}{n-m}\mathbf{H}_{\mathcal{D}_r}^{-1}\nabla\mathcal{L}(\mathcal{D}_r, \boldsymbol{w}^*),$$

and then injecting noise (commonly Gaussian) to the updated model, which is calibrated with respect to an upper bound on the norm difference between $\boldsymbol{w}_r^*$ and $\boldsymbol{w}_r$. The details on certified unlearning mechanisms utilizing second-order Newton updates are given under Appendix B.1.

**This work.** In contrast to the conventional certified unlearning problem, in our scenario the unlearning mechanism only has access to the surrogate dataset $\mathcal{D}_s$, as opposed to the source dataset $\mathcal{D}$. Our goal is then to develop a certified unlearning mechanism with only access to $\mathcal{D}_s$. In the next section, we present a novel approach to address this challenge, where we propose a novel Gaussian mechanism building on a second-order Newton update, where the noise is calibrated as a function of the statistical distance between the source and surrogate data distributions, without having access to the original training set. In scenarios where the statistical distance is not readily available, we demonstrate a simple methodology to estimate this by using the model. While providing strong theoretical guarantees, our approach is versatile and can be applied to a wide range of unlearning algorithms that use a single-step second-order Newton update as the approximation method.

## 4. Methodology

Our approach consists of the following key steps:

1. **(Hessian estimation.)** Our approach builds on second-order unlearning, which requires the Hessian of the source dataset to update the model for forgetting. As we do not have access to the source data, we estimate the true Hessian $\mathbf{H}_{\mathcal{D}}$ using the Hessian of the surrogate dataset $\mathbf{H}_{\mathcal{D}_s}$. Using the surrogate Hessian, we estimate the true Hessian $\mathbf{H}_{\mathcal{D}_r}$ of the retain samples $\mathcal{D}_r$.
2. **(Model update.)** Using the estimated Hessian $\widehat{\mathbf{H}}_{\mathcal{D}_r}$ of the retain samples, we update the model $\boldsymbol{w}^*$ (trained on $\mathcal{D}$) using a single-step second-order Newton update.
3. **(Noise calibration.)** Finally, we employ a Gaussian mechanism adding noise $\boldsymbol{n}$ to the updated model $\widehat{\boldsymbol{w}}_r$. To ensure certified unlearning, we calibrate the noise using an upper bound on the $L_2$ norm distance between the estimated model $\widehat{\boldsymbol{w}}_r$ and the true unlearned model $\boldsymbol{w}_r^*$, along with the total variation distance $\mathrm{TV}(\rho \parallel \nu)$ between the source and surrogate datasets.

Before we describe the details of these steps, we first provide a useful technical assumption.

**Assumption 4.1.** The loss function $\mathcal{L}$ used during the training of the model parameters is $L$-Lipschitz, $\alpha$-strongly convex, $\beta$-smooth, and $\gamma$-Hessian Lipschitz.

Details of these assumptions are provided in Appendix A. We next describe our individual steps.

**1. Hessian estimation.** Our mechanism approximates the model retrained from scratch, i.e., trained only on the retained samples of the source dataset, by using the surrogate dataset and a one-step second-order Newton update.

The second-order Newton update is the product of the inverse Hessian and the gradient vector, both evaluated at $\boldsymbol{w}^*$, the model trained on the training dataset $\mathcal{D}$. If the original retain data $\mathcal{D}_r$ was available, the update would be, $\boldsymbol{w}_r = \boldsymbol{w}^* - \mathbf{H}_{\mathcal{D}_r}^{-1}\nabla\mathcal{L}(\mathcal{D}_r, \boldsymbol{w}^*)$. Since $\mathcal{D}_r$ is unavailable, we approximate its Hessian as

$$\widehat{\mathbf{H}}_{\mathcal{D}_r} = \frac{n\mathbf{H}_{\mathcal{D}_s} - m\mathbf{H}_{\mathcal{D}_u}}{n-m}. \tag{1}$$

**2. Model update.** Using the estimated Hessian $\widehat{\mathbf{H}}_{\mathcal{D}_r}$, we then update the model. The update also requires $\nabla\mathcal{L}(\mathcal{D}_r, \boldsymbol{w}^*)$, which we express using the fact that $\nabla\mathcal{L}(\mathcal{D}, \boldsymbol{w}^*) = 0$ for the fully trained model and therefore,

$$\nabla\mathcal{L}(\mathcal{D}_r, \boldsymbol{w}^*) = \frac{-m\nabla\mathcal{L}(\mathcal{D}_u, \boldsymbol{w}^*)}{n-m}. \tag{2}$$

Substituting (1) and (2) into the second-order Newton update yields our model update for unlearning,

$$\widehat{\boldsymbol{w}}_r = \boldsymbol{w}^* + \frac{m}{n-m}\widehat{\mathbf{H}}_{\mathcal{D}_r}^{-1}\nabla\mathcal{L}(\mathcal{D}_u, \boldsymbol{w}^*). \tag{3}$$

**3. Noise calibration.** To ensure certified unlearning, we then introduce a Gaussian mechanism with the noise scaled according to: 1) an upper bound on $\|\boldsymbol{w}_r^* - \widehat{\boldsymbol{w}}_r\|_2$, 2) a fidelity criterion based on the statistical distance between the source and surrogate data distributions. Specifically, the final model is given by,

$$\widehat{\boldsymbol{w}}_r' := \widehat{\boldsymbol{w}}_r + \boldsymbol{n}$$

where $\boldsymbol{n} \sim \mathcal{N}(0, \sigma^2\mathbf{I})$ such that,

$$\sigma = \frac{\Delta}{\epsilon}\sqrt{2\ln(1.25/\delta)} \tag{4}$$

and

$$\|\boldsymbol{w}_r^* - \widehat{\boldsymbol{w}}_r\|_2 \le \Delta$$
$$\triangleq \frac{2\gamma L m^2}{\alpha^3 n_1^2} + \Bigg(\|\nabla\mathcal{L}(\mathcal{D}_u, \boldsymbol{w}^*)\|_2$$
$$\frac{m(n_1 - n_2)\beta + 2mn_2\beta\mathrm{TV}(\rho \parallel \nu)}{(n_1\alpha - m\beta)(n_2\alpha - m\beta)}\Bigg) \tag{5}$$

to achieve $(\epsilon, \delta)$-certified unlearning. Algorithm 1 presents the individual steps for our certified unlearning mechanism $\widehat{\mathcal{U}}$. We next provide the theoretical justification behind (4) and (5) in Theorem 4.2 and Theorem 4.3.

**Algorithm 1** Unlearning Mechanism Leveraging Surrogate Data Statistics

---

**Require:** Unlearning dataset $\mathcal{D}_u$, trained model parameters $\boldsymbol{w}^*$ (from $\mathcal{A}(\mathcal{D})$), data statistics $\mathcal{S}(\mathcal{D}_s) : \mathbf{H}_{\mathcal{D}_s}$, upper bound $\Delta$, privacy parameters $\epsilon, \delta$

**Ensure:** Updated model parameters $\widehat{\boldsymbol{w}}_r'$ after unlearning

---

1: Compute $\sigma = \frac{\Delta}{\epsilon}\sqrt{2\ln(1.25/\delta)}$
2: Compute $\widehat{\mathbf{H}}_{\mathcal{D}_r} = \frac{n\mathbf{H}_{\mathcal{D}_s} - m\mathbf{H}_{\mathcal{D}_u}}{n-m}$
3: Update $\widehat{\boldsymbol{w}}_r = \boldsymbol{w}^* + \frac{m}{n-m}\widehat{\mathbf{H}}_{\mathcal{D}_r}^{-1}\nabla\mathcal{L}(\boldsymbol{w}^*, \mathcal{D}_u)$
4: Sample $\boldsymbol{n} \sim \mathcal{N}(0, \sigma^2\mathbf{I})$
5: **Return** $\widehat{\boldsymbol{w}}_r' := \widehat{\boldsymbol{w}}_r + \boldsymbol{n}$

---

## 4.1. Theoretical Principles

In this section we provide the theoretical intuition behind our mechanism. In Theorem 4.2, we derive an upper bound on the difference between the true retrained model $\boldsymbol{w}_r^*$, trained from scratch on the retain data $\mathcal{D}_r$, and the approximate model $\widehat{\boldsymbol{w}}_r$, which uses the surrogate data $\mathcal{D}_s$. This bound is formulated in terms of the total variation distance $\mathrm{TV}(\rho \parallel \nu)$ between the source and surrogate data distributions.

**Theorem 4.2.** *Consider a loss function $\mathcal{L}$ satisfying Assumption 4.1, and a surrogate dataset $\mathcal{D}_s$ with $n_2$ samples drawn from a distribution $\nu$, to mimic the source dataset $\mathcal{D}$ with $n_1$ drawn from a distribution $\rho$, over the support set $\mathcal{X} \times \mathcal{Y}$. Define the retrained model over the retain samples as $\boldsymbol{w}_r^*$ and the model achieved after unlearning as $\widehat{\boldsymbol{w}}_r$. Also, assume that $n_1$ and $n_2$ are sufficiently large and $n_1, n_2 \geq \frac{m\beta}{\alpha}$. Then, the following upper bound holds,*

$$\|\boldsymbol{w}_r^* - \widehat{\boldsymbol{w}}_r\|_2 \leq \Delta$$

*where $\Delta$ is as defined in (5).*

*Proof.* The proof is provided in Appendix C. □

The next theorem presents our certified unlearning guarantees under a given privacy budget $\epsilon$ and confidence $\delta$ when noise scaled by $\Delta$ is added to the approximate model in (3).

**Theorem 4.3.** *Consider a dataset $\mathcal{D}$ where data samples follow the distribution $\rho$, and a surrogate dataset $\mathcal{D}_s$ where data samples follow the distribution $\nu$. Given a forget set $\mathcal{D}_u \subseteq \mathcal{D}$, and the hypothesis set $\mathcal{H}$, the unlearning mechanism $\widehat{\mathcal{U}}$ satisfies $(\epsilon, \delta)$-certified unlearning.*

*Proof.* The proof is provided in Appendix C.1. □

Thus, when $\mathrm{TV}(\rho \parallel \nu)$ is large, the noise magnitude $\sigma$ increases, ensuring certified guarantees even when the surrogate distribution significantly differs from the source.

A key challenge is estimating (or upper bounding) $\mathrm{TV}(\rho \parallel \nu)$ without access to $\mathcal{D}$. In the next section, we introduce a heuristic method to approximate this distance (or an upper bound) using only $\mathcal{D}_s$ and the trained model $\boldsymbol{w}^*$. This enables the implementation of Algorithm 1 without direct access to $\mathcal{D}$, which is crucial for privacy-sensitive applications and real-world deployments.

## 4.2. From Theory to Practice

In this section, we first propose an upper bound using Kullback-Leibler (KL) divergence. While total variation distance would be preferable, we use KL for efficiency due to no access to $\mathcal{D}$. Next, we approximate KL without direct access to exact samples by training a model on $\mathcal{D}_s$ and utilizing models as conditional probabilities. We sample from input marginal distribution using energy-based modeling to compute the KL. Finally, we estimate the KL between input marginal distributions using the Donsker-Varadhan variational representation (Donsker & Varadhan, 1983). The details of these steps are explained below.

To apply the bound in Theorem 4.2, we require the exact total variation distance $\mathrm{TV}(\rho \parallel \nu)$ or an upper bound. In practice, we approximate this bound using the KL divergence, leveraging heuristics outlined in this section. In Corollary 4.4, we provide an upper bound based on the KL divergence between surrogate and source data distributions.

**Corollary 4.4.** *Under the same assumptions and definitions in Theorem 4.2, the following upper bound holds:*

$$\|\boldsymbol{w}_r^* - \widehat{\boldsymbol{w}}_r\|_2 \leq \frac{2\gamma L m^2}{\alpha^3 n_1^2} + \left(\|\nabla\mathcal{L}(\mathcal{D}_u, \boldsymbol{w}^*)\|_2\right.$$
$$\left.\frac{m(n_1 - n_2)\beta + 2mn_2\beta\sqrt{1 - \exp(-\mathrm{KL}(\nu \parallel \rho))}}{(n_1\alpha - m\beta)(n_2\alpha - m\beta)}\right)$$

*Proof.* The proof is available in Appendix D.1. □

To approximate $\mathrm{KL}(\nu \parallel \rho)$, we leverage the model $\boldsymbol{w}^*$ trained on the entire dataset $\mathcal{D}$. Let $f(\boldsymbol{w}, \boldsymbol{x})$ denote the probability simplex over classes parameterized by the model $\boldsymbol{w}$ for a sample $\boldsymbol{x}$, with $f(\boldsymbol{w}, \boldsymbol{x})_y$ representing the probability of class $y$. Assuming $\tilde{\boldsymbol{w}}^*$ is the model trained on the surrogate dataset $\mathcal{D}_s$, the KL divergence can be decomposed as shown in Proposition 4.5.

**Proposition 4.5.** *Let $f(\boldsymbol{w}, \boldsymbol{x})$ output a probability simplex over classes for a data sample $\boldsymbol{x}$, parameterized by $\boldsymbol{w}$. Given trained models $\boldsymbol{w}^*$ and $\tilde{\boldsymbol{w}}^*$, such that $\boldsymbol{w}^*$ is trained on $\mathcal{D}$ and $\tilde{\boldsymbol{w}}^*$ on $\mathcal{D}_s$, where data samples from $\mathcal{D}$ and $\mathcal{D}_s$ follow distributions $\rho$ and $\nu$, the KL divergence $\mathrm{KL}(\nu \parallel \rho)$ can be decomposed as,*

$$\mathrm{KL}(\nu \parallel \rho) \approx \frac{1}{n}\sum_{(\boldsymbol{x},y)\in\mathcal{D}_s}\log\frac{f(\tilde{\boldsymbol{w}}^*, \boldsymbol{x})_y}{f(\boldsymbol{w}^*, \boldsymbol{x})_y}$$
$$+ \mathrm{KL}(\nu(\boldsymbol{x}) \parallel \rho(\boldsymbol{x}))$$

*Proof.* The derivation is given in Appendix D.2. □

Proposition 4.5 decomposes the KL divergence into two components: (1) divergence between conditional distributions, which we can approximate using the classifiers, and (2) divergence between input marginal distributions.

To estimate the latter, we leverage the hidden energy-based model in the classifier $\boldsymbol{w}^*$ to sample from the true input marginal distribution $\rho(\boldsymbol{x})$. These samples, combined with the surrogate dataset, enable us to approximate the KL divergence between marginal distributions. The next section details the technical steps for sampling from $\rho(\boldsymbol{x})$ using only the trained model parameters.

***Sampling from input marginal distribution.*** Inspired by (Grathwohl et al., 2019), we leverage the implicit energy-based model of the trained model $\boldsymbol{w}^*$ to sample from the approximated input marginal distribution $\hat{\rho}(\boldsymbol{x})$ given by:

$$\hat{\rho}(\boldsymbol{x}) = \frac{\exp(-E(\boldsymbol{x}))}{Z}$$

where the energy function is defined as $E(\boldsymbol{x}) = -\log \sum_{y \in \mathcal{Y}} \exp(f(\boldsymbol{w}^*, \boldsymbol{x})_y)$. Here, $f(\boldsymbol{w}^*, \boldsymbol{x})_y$ denotes the logit score for label $y$ under $\boldsymbol{w}^*$, and the summation runs over the label space $\mathcal{Y}$.

To sample from $\hat{\rho}(\boldsymbol{x})$, we employ Stochastic Gradient Langevin Dynamics (SGLD), which iteratively refines samples without explicitly computing the normalization constant $Z$. These samples, combined with the surrogate data, allow us to approximate the input marginal KL divergence.

In the next section, we present how to estimate this divergence using a variational representation, ensuring a practical approach for our unlearning mechanism. Further details on the energy-based modeling, SGLD sampling procedure, and convergence criteria can be found in Appendix D.3.

***Approximating KL Distance Between Input Marginal Distributions.*** After generating samples from the approximated source distribution $\hat{\rho}(\boldsymbol{x})$ using Langevin dynamics, we approximate the KL divergence between the surrogate distribution $\nu$ and the approximated source distribution $\hat{\rho}$ by leveraging the Donsker-Varadhan variational representation,

$$\text{KL}(\nu(\boldsymbol{x}) \parallel \hat{\rho}(\boldsymbol{x})) = \sup_T \mathbb{E}_{X \sim \nu} [T(X)] \\ - \log \mathbb{E}_{X \sim \hat{\rho}} [\exp(T(X))] \quad (6)$$

where $T$ is a variational function, parametrized by a neural network, that maps input samples to real-valued scores.

To approximate the expectations in (6), we rely on the samples generated through Langevin dynamics. Given $k$ samples sampled from $\hat{\rho}(\boldsymbol{x})$, forming a set $\{\hat{\boldsymbol{x}}_i\}_{i=1}^k$, and $n$ samples from $\nu(\boldsymbol{x})$ forming the set $\{\boldsymbol{x}_i\}_{i=1}^k$, the KL divergence

can be approximated as,

$$\text{KL}(\nu(\boldsymbol{x}) \parallel \hat{\rho}(\boldsymbol{x})) \approx \sup_T \frac{1}{n} \sum_{i=1}^n T(\boldsymbol{x}_i) \\ - \log \left( \frac{1}{k} \sum_{j=1}^k \exp(T(\hat{\boldsymbol{x}}_j)) \right)$$

Finally, building on this result, we refine the decomposition of the KL divergence from Proposition 4.5 to explicitly account for the approximation of $\text{KL}(\nu \parallel \rho)$ as follows.

**Proposition 4.6.** *Consider the setting from Proposition 4.5 and assume $\text{KL}(\nu \parallel \rho)$ is approximated using sampling from $\hat{\rho}(\boldsymbol{x})$ and the Donsker-Varadhan variational representation described in (6). The $\text{KL}(\nu \parallel \rho)$ can then be expressed as,*

$$\text{KL}(\nu \parallel \rho) \approx \frac{1}{n} \sum_{(\boldsymbol{x}, y) \in \mathcal{D}_s} \log \frac{f(\tilde{\boldsymbol{w}}^*, \boldsymbol{x})_y}{f(\boldsymbol{w}^*, \boldsymbol{x})_y} \\ + \sup_T \left( \frac{1}{n} \sum_{i=1}^n T(\boldsymbol{x}_i) - \log \left( \frac{1}{k} \sum_{j=1}^k \exp(T(\hat{\boldsymbol{x}}_j)) \right) \right)$$

In our experiments, we demonstrate our evaluations with both synthetic and real-world datasets. For the latter, SGLD sampling and Proposition 4.6 will be instrumental in our experiments with real-world datasets.

## 5. Experiments

We systematically evaluate our approach on synthetic and real-world datasets to demonstrate its effectiveness in achieving certified unlearning. Unless otherwise noted, we adopt a linear training model with forget ratio of $0.1$, and an $L_2$ regularization constant $\lambda = 0.01$. The loss function is assumed to be $\alpha$-strongly convex, $L$-Lipschitz, $\beta$-smooth, and $\gamma$-Hessian Lipschitz. In line with prior works (Koh & Liang, 2017; Wu et al., 2023b;a; Zhang et al., 2024) we tune $\alpha$, $L$, $\beta$, and $\gamma$ for each experimental setting, which preserves theoretical soundness but may lead to approximate certifications. All details about the parameter study and implementation are given under Appendix E.

We now turn to our empirical evaluation, where we assess the effectiveness of our approach through a series of experiments. We first provide explanations about the evaluation metrics used to evaluate our unlearning mechanism. Then, we provide synthetic and real-world dataset experiments. In addition to that, we also investigate our methodology over different setups to demonstrate its effectiveness.

***Performance Metrics.*** We evaluate the performance using train, test, retain, and forget accuracies on their respective data splits. Additionally, we employ the unlearning-specific membership inference attack (MIA) (Kurmanji et al., 2023),

| $\zeta$ | Method | Train | Test | Retain | Forget | MIA | RT |
|---|---|---|---|---|---|---|---|
| – | Retrain | 77.0 % | 72.0 % | 77.4 % | 73.6 % | 47.63 % | 10 |
| – | Unlearn (+) | 77.1 % | 72.6 % | 77.5 % | 74.1 % | 47.63 % | 10 |
| 0.02 | **Unlearn (-)** | **77.2 %** | **72.2 %** | **77.5 %** | **74.8 %** | **48.89 %** | **7** |
| 0.04 | **Unlearn (-)** | **77.3 %** | **72.4 %** | **77.6 %** | **74.4 %** | **48.89 %** | **10** |
| 0.06 | **Unlearn (-)** | **77.3 %** | **72.2 %** | **77.6 %** | **74.3 %** | **48.37 %** | **10** |
| 0.08 | **Unlearn (-)** | **77.3 %** | **72.4 %** | **77.6 %** | **74.4 %** | **48.15 %** | **10** |
| 0.1 | **Unlearn (-)** | **77.3 %** | **72.7 %** | **77.7 %** | **74.1 %** | **48.30 %** | **10** |

*Table 1.* Evaluation of unlearning performance while varying the off-diagonal elements ($\zeta$) of the unit covariance.

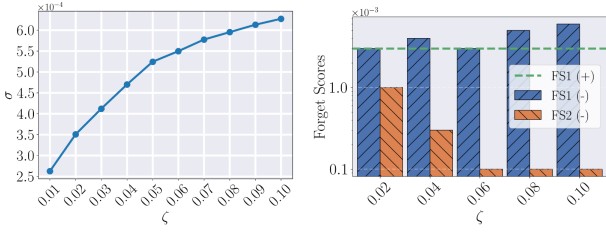

(a) Required noise variance $\sigma$      (b) Forget scores

*Figure 1.* **(a):** Required variance $\sigma$ for achieving certified unlearning on synthetic datasets as a function of the off-diagonal elements ($\zeta$). **(b):** Forget scores achieved for synthetic datasets.

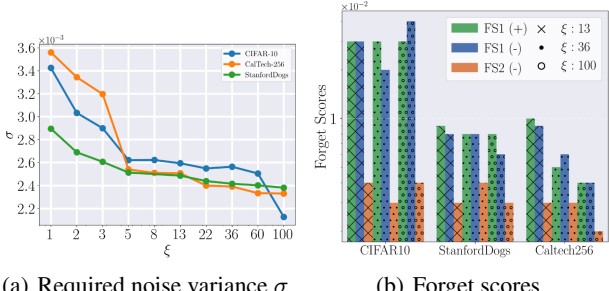

(a) Required noise variance $\sigma$      (b) Forget scores

*Figure 2.* **(a):** Required variance $\sigma$ for achieving certified unlearning across CIFAR10, StanfordDogs, and Caltech256 datasets as a function of the concentration parameter $\xi$. **(b):** Forget scores achieved for CIFAR10, StanfordDogs, and Caltech256.

where an accuracy of 50% means the attack can not distinguish whether a specific sample belongs to the forget or test dataset. Relearn time (RT) (Golatkar et al., 2020) measures how many additional training iterations are required to restore the model's performance on the forgotten data after it is reintroduced. Intuitively, the unlearned model should give high relearn time scores which indicates the model effectively unlearns the forget dataset. Finally, we report the forget score (FS) (Triantafillou et al., 2024), which quantifies how closely the predictions of the unlearned model align with those of a model retrained from scratch. A higher forget score indicates stronger unlearning and higher indistinguishability between retrained and unlearned models.

Overall, we denote our method as "Unlearn (-)" indicating no access to the statistical information about the source data, unlearning method utilizing statistical information about the source data as "Unlearn (+)" and the model retrained from scratch over the retain data as "Retrain". Also, we report three FS variants: "FS1 (+)" applies the noise required by Unlearn (+), "FS1 (-)" applies the noise required by our proposed Unlearn (-), and "FS2 (-)" applies the noise from Unlearn (+) to Unlearn (-). Comparing these scores demonstrates that our proposed approach is required to achieve certified unlearning while utilizing a surrogate dataset.

***Synthetic Experiments.*** We generate an source dataset of 15000 samples from a 50-dimensional standard Gaussian, $\mathcal{N}(\mathbf{0}, \mathbf{I})$. A corresponding surrogate dataset of the same size is drawn from $\mathcal{N}(\mathbf{0}, \zeta\mathbf{1} - (\zeta + 1)\mathbf{I})$, where $\zeta \in [0.01, 0.1]$ controls the off-diagonal covariance terms. Varying $\zeta$ modulates the KL divergence between the source and surrogate distributions, influencing the noise variance needed for certified unlearning. As demonstrated in Figure 1(a), the required noise variance is increased following Theorem 4.2.

Table 1 reports train, test, retain, and forget accuracies alongside the MIA score and RT. Despite the required noise increasing with larger KL divergence, our method Unlearn (-) achieves utility comparable to other methods. These results underscore that appropriately scaling noise according to distributional distance can preserve model performance while guaranteeing unlearning. From the forget scores given in Figure 1(b), we observe that while FS1 (+) and FS1 (-) can achieve similar forget scores, FS2 (-) is always lower than

the others, implying that to achieve similar certification with Unlearn (+), our proposed noise is required. We report additional experiments using different random seeds along with corresponding error bars in Appendix F.

***Real-World Dataset Experiments.*** We further evaluate our method on CIFAR10 (Krizhevsky et al., 2009), Caltech256 (Griffin et al., 2007), and StanfordDogs (Khosla et al., 2011), by dividing each dataset into an source and a surrogate subset according to a Dirichlet distribution with concentration $\xi$. Lower values of $\xi$ lead to more skewed class splits and thus greater distributional divergences. We show these results in Figure 2(a). We observe that the required noise variance decreases while increasing the concentration parameter $\xi$. We approximate the KL distance between source and surrogate datasets without accessing source data by using Proposition 4.6. We use embeddings from a ResNet18 (He et al., 2016) model, following (Guo et al., 2019).

In Table 2 we report the train, test, retain, and forget accuracies for all datasets. Our method Unlearn (-) achieves comparable accuracy over all data splits similar to other methods while utilizing only the surrogate datasets. We also report the MIA and RT metrics in Table 3 showing that our unlearning performance is close to the other methods. Finally, in Figure 2(b) we demonstrate the forget scores for

| ξ | Method | CIFAR-10 | | | | StanfordDogs | | | | Caltech256 | | | |
|---|--------|-------|------|--------|--------|-------|------|--------|--------|-------|------|--------|--------|
| | | Train | Test | Retain | Forget | Train | Test | Retain | Forget | Train | Test | Retain | Forget |
| | Retrain | 77.6 % | 76.2 % | 77.8 % | 76.0 % | 86.1 % | 73.7 % | 87.3 % | 75.2 % | 87.1 % | 72.0 % | 88.8 % | 72.2 % |
| 13 | Unlearn (+) | 77.9 % | 76.4 % | 78.0 % | 76.3 % | 84.1 % | 71.9 % | 85.3 % | 73.2 % | 86.8 % | 70.8 % | 88.6 % | 71.2 % |
| | **Unlearn (-)** | **77.5 %** | **76.1 %** | **77.7 %** | **75.8 %** | **84.0 %** | **72.2 %** | **85.2 %** | **73.1 %** | **87.0 %** | **71.5 %** | **88.4 %** | **74.6 %** |
| | Retrain | 78.0 % | 76.7 % | 78.2 % | 76.5 % | 84.6 % | 75.1 % | 86.0 % | 71.4 % | 84.9 % | 74.6 % | 86.2 % | 73.3 % |
| 36 | Unlearn (+) | 77.4 % | 76.5 % | 77.6 % | 75.7 % | 84.5 % | 75.6 % | 85.9 % | 71.7 % | 84.7 % | 73.2 % | 86.0 % | 73.8 % |
| | **Unlearn (-)** | **77.3 %** | **76.4 %** | **77.5 %** | **75.7 %** | **84.4 %** | **75.7 %** | **85.8 %** | **72.0 %** | **84.9 %** | **73.5 %** | **86.0 %** | **74.9 %** |
| | Retrain | 78.0 % | 76.9 % | 78.0 % | 77.8 % | 82.9 % | 76.0 % | 84.2 % | 71.1 % | 83.5 % | 74.1 % | 84.7 % | 73.3 % |
| 100 | Unlearn (+) | 78.2 % | 77.3 % | 78.3 % | 77.2 % | 83.8 % | 75.7 % | 85.1 % | 72.2 % | 82.1 % | 73.0 % | 83.2 % | 72.8 % |
| | **Unlearn (-)** | **78.1 %** | **77.2 %** | **78.1 %** | **77.3 %** | **83.7 %** | **75.6 %** | **85.0 %** | **72.0 %** | **82.0 %** | **72.8 %** | **83.0 %** | **72.9 %** |

*Table 2.* Train, test, retain, forget set accuracies for each method across CIFAR10, StanfordDogs, and Caltech256 datasets while varying the concentration parameter ($\xi$) of the Dirichlet distribution.

| ξ | Method | CIFAR-10 | | StanfordDogs | | Caltech256 | |
|---|--------|------|----|------|----|------|----|
| | | MIA | RT | MIA | RT | MIA | RT |
| | Retrain | 51.14 % | 14 | 51.95 % | 70 | 50.97 % | 20 |
| 13 | Unlearn (+) | 52.68 % | 10 | 51.61 % | 15 | 51.20 % | 20 |
| | **Unlearn (-)** | **52.59 %** | **23** | **51.49 %** | **15** | **52.00 %** | **16** |
| | Retrain | 49.97 % | 2 | 50.87 % | 20 | 50.21 % | 21 |
| 36 | Unlearn (+) | 50.15 % | 10 | 50.05 % | 18 | 47.39 % | 21 |
| | **Unlearn (-)** | **49.80 %** | **6** | **50.14 %** | **19** | **50.46 %** | **17** |
| | Retrain | 49.76 % | 7 | 52.49 % | 19 | 48.01 % | 17 |
| 100 | Unlearn (+) | 48.90 % | 13 | 52.02 % | 21 | 52.05 % | 16 |
| | **Unlearn (-)** | **48.83 %** | **32** | **51.94 %** | **16** | **52.33 %** | **14** |

*Table 3.* MIA and RT metrics given for each method across CIFAR10, StanfordDogs, and Caltech256 datasets while varying the concentration parameter ($\xi$) of the Dirichlet distribution.

| FR | Method | Train | Test | Retain | Forget | MIA | RT |
|----|--------|-------|------|--------|--------|-----|----|
| | Retrain | 87.1 % | 73.7 % | 87.2 % | 73.8 % | 52.1 % | 10 |
| 0.01 | Unlearn (+) | 87.3 % | 74.1 % | 87.3 % | 74.5 % | 53.2 % | 10 |
| | **Unlearn (-)** | **87.1 %** | **74.1 %** | **87.2 %** | **74.1 %** | **53.1 %** | **10** |
| | Retrain | 82.9 % | 76.0 % | 84.2 % | 71.1 % | 52.5 % | 19 |
| 0.1 | Unlearn (+) | 83.8 % | 75.7 % | 85.1 % | 72.2 % | 52.0 % | 21 |
| | **Unlearn (-)** | **83.7 %** | **75.6 %** | **85.0 %** | **72.0 %** | **51.9 %** | **16** |
| | Retrain | 85.6 % | 72.4 % | 88.7 % | 73.3 % | 50.6 % | 40 |
| 0.2 | Unlearn (+) | 84.9 % | 71.8 % | 88.3 % | 71.5 % | 51.8 % | 40 |
| | **Unlearn (-)** | **85.0 %** | **71.4 %** | **88.0 %** | **72.6 %** | **52.0 %** | **40** |

*Table 4.* Evaluation of unlearning performance across varying forget ratios (FR) on StanfordDogs dataset with $\xi = 100$.

all datasets and selected concentration parameters. This implies the necessity of our noise scaling approach while using a surrogate dataset to achieve certified unlearning.

Additional experiments with different random seeds, corresponding error bars, and evaluations of our heuristic KL approximation—used for noise calibration without accessing the source data—against KL estimates via the Donsker-Varadhan method (with data access) are reported in Appendix F, highlighting the gap between practical estimation and the exact quantity required for certified unlearning.

***Experiments with Different Forget Ratios.*** We conducted extensive experiments on the StanfordDogs dataset with varying forget ratios to assess how forget ratio impacts unlearning. The results in Table 4 show that our method Unlearn (-) scales well across different forget set ratios. Also, results under the MIA and RT columns indicate that similar unlearning performance is achieved across different forget ratios with Unlearn (+) and Retrain models. These findings confirm the robustness of our approach.

***Mixed-Linear Network Experiments.*** While the convexity and smoothness assumptions in Assumption 4.1 may not hold for general neural networks, there exist practical architectures that satisfy these conditions while retaining strong utility. To this end, we adopt the mixed-linear networks (Golatkar et al., 2021), which linearizes a pre-trained neural

network using a first-order Taylor expansion. Specifically, the network output is approximated via its Neural Tangent Kernel (Jacot et al., 2018) formulation, transforming the objective into a convex optimization problem. This approximation allows for efficient and tractable unlearning while preserving much of the model's predictive performance.

In Table 5, we report results on CIFAR-10 under two settings using this architecture. One with randomly selecting 10% of the data as forget set and the other with removing all samples belonging to class 0. In both cases, our method achieves effective certified unlearning and maintains competitive accuracy on the retained data, demonstrating that mixed linear networks provide a practical and theoretically sound foundation for unlearning in neural models. MIA

| – | Method | Train | Test | Retain | Forget | MIA | RT |
|----|--------|-------|------|--------|--------|-----|----|
| | Retrain | 93.6 % | 86.4 % | 95.6 % | 84.7 % | 51.2 % | 53 |
| 0.1 | Unlearn (+) | 93.7 % | 86.4 % | 94.8 % | 87.2 % | 51.3 % | 54 |
| | **Unlearn (-)** | **94.1 %** | **85.2 %** | **94.9 %** | **86.8 %** | **52.1 %** | **54** |
| | Retrain | 81.7 % | 72.3 % | 92.7 % | 0 % | – | 142 |
| 0 | Unlearn (+) | 82.2 % | 72.5 % | 93.2 % | 4.2 % | – | 135 |
| | **Unlearn (-)** | **82.4 %** | **72.4 %** | **93.5 %** | **5.1 %** | **–** | **132** |

*Table 5.* Evaluation of unlearning performance on CIFAR-10 using mixed-linear networks with $\xi = 100$. In this table, "0.1" indicates that 10% of the data is used as the forget set, and "0" denotes the class selected for unlearning.

| Arch | Method | Train | Test | Retain | Forget | MIA | RT |
|---|---|---|---|---|---|---|---|
| | Retrain | 78.0 % | 76.9 % | 78.0 % | 77.8 % | 49.76 % | 7 |
| L | Unlearn (+) | 78.2 % | 77.3 % | 78.3 % | 77.2 % | 48.90 % | 13 |
| | **Unlearn (-)** | **78.1 %** | **77.2 %** | **78.1 %** | **77.3 %** | **48.83 %** | **32** |
| | Retrain | 81.6 % | 79.8 % | 82.1 % | 78.4 % | 49.94 % | 40 |
| C+L | Unlearn (+) | 80.8 % | 78.4 % | 81.3 % | 78.1 % | 51.32 % | 45 |
| | **Unlearn (-)** | **80.5 %** | **78.1 %** | **80.9 %** | **77.5 %** | **50.71 %** | **46** |
| | Retrain | 83.0 % | 80.3 % | 83.1 % | 81.2 % | 50.86 % | 22 |
| 2C+L | Unlearn (+) | 84.3 % | 81.4 % | 84.3 % | 81.7 % | 51.28 % | 20 |
| | **Unlearn (-)** | **82.9 %** | **80.5 %** | **83.1 %** | **81.1 %** | **50.05 %** | **22** |

*Table 6.* Evaluation of unlearning performance across different model architectures: a single linear layer (L), a convolutional layer followed by a linear layer (C+L), and two convolutional layers followed by a linear layer (2C+Lin).

scores are omitted for class unlearning because the attack is designed to distinguish between test and forget samples; forgetting an entire class greatly increases distinguishability, making the MIA score uninformative.

***Unlearning Across Model Architectures.*** To evaluate the generality of our approach, we train three different architectures on CIFAR-10 using a Dirichlet concentration of $\xi=100$: a single linear layer ("L"), a convolutional layer followed by a linear layer ("C+L"), and two convolutional layers with a linear layer ("2C+L"). As shown in Table 6, our method maintains accuracy comparable to others, while keeping the MIA score close to 50%. Also, the RT metric implies that the unlearning succeeded in removing the influence of the forget samples. Finally, for the C+L architecture, the observed FS1 (+), FS1 (-), and FS2 (-) values are 0.08, 0.08, and 0.05, respectively, while for 2C+L, they are lower at 0.04, 0.04, and 0.02. These results reinforce that the introduced noise is essential for the unlearning process.

***MNIST-USPS Experiment.*** To illustrate our contribution in a practical setting, we consider MNIST (Lecun et al., 1998) and USPS (Hull, 1994) datasets and analyze the following cases. First, we train a model on MNIST and apply our unlearning framework by selecting a random forget set from MNIST while using USPS as the surrogate dataset ($M \rightarrow U$). Second, we reverse the process, training a model on USPS and unlearning with MNIST as the surrogate ($U \rightarrow M$). As can be seen from Table 7, in both cases the unlearning performance of our method Unlearn (-) is similar to the other methods we are comparing with.

| Task | Method | Train | Test | Retain | Forget | MIA | RT |
|---|---|---|---|---|---|---|---|
| | Retrain | 94.2 % | 91.4 % | 94.3 % | 92.5 % | 51.23 % | 11 |
| M→U | Unlearn (+) | 94.1 % | 91.3 % | 94.1 % | 90.7 % | 50.15 % | 13 |
| | **Unlearn (-)** | **94.1 %** | **91.1 %** | **94.1 %** | **91.5 %** | **50.54 %** | **13** |
| | Retrain | 95.2 % | 91.1 % | 95.1 % | 92.9 % | 50.71 % | 21 |
| U→M | Unlearn (+) | 93.7 % | 91.3 % | 95.3 % | 91.7 % | 51.93 % | 24 |
| | **Unlearn (-)** | **93.5 %** | **90.4 %** | **94.9 %** | **90.9 %** | **50.60 %** | **23** |

*Table 7.* Evaluation of unlearning performance with MNIST and USPS dataset experiments.

# 6. Conclusion

We introduce a certified unlearning framework that enables data removal without requiring access to the original training data statistics. Unlike existing methods, our approach utilizes a surrogate dataset and calibrates noise based on statistical distance, ensuring provable guarantees. We establish theoretical bounds, develop a practical noise-scaling mechanism, and validate our method through experiments on synthetic and real-world datasets. Our results demonstrate certified unlearning can be achieved by utilizing a surrogate dataset while maintaining utility and privacy guarantees.

## Software

Our main implementation used for this paper is available at `https://github.com/info-ucr/certified-unlearning-surr-data`. We also implemented the mixed-linear networks (Golatkar et al., 2021) from scratch, the code is available at `https://github.com/info-ucr/mixed-privacy-forgetting`.

## Acknowledgements

This work was supported in part by the NSF CAREER Award CCF-2144927, NSF Award CCF-2008020, DURIP N000141812252, the UCR OASIS Fellowship, and the Amazon Research Award.

## Impact Statement

Unlearning is increasingly critical due to evolving privacy regulations, such as GDPR, CCPA and CPPA, which mandate mechanisms to effectively erase private or sensitive data from trained machine learning models. Retraining these models from scratch to remove specific data points is computationally infeasible. Traditional unlearning methods circumvent exhaustive retraining but typically require full access to the original source data, an assumption often unrealistic in practical scenarios due to privacy concerns, storage limitations, or regulatory restrictions on data retention. Our work directly addresses this crucial gap by proposing a certified unlearning framework that does not rely on the availability of original training data. Instead, we leverage surrogate datasets that approximate the original data distribution to guide the unlearning process. By carefully calibrating noise injection based on statistical distances between original and surrogate datasets, our method ensures rigorous theoretical guarantees on unlearning performance, thereby providing a principled alternative to heuristic methods. This approach significantly broadens the practical applicability of certified unlearning methods, ensuring compliance with privacy requirements even when access to original training data is restricted or completely unavailable.

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

# A. Assumptions

For the loss function $\mathcal{L}$ used to train the model parameters, we have the following assumptions listed in Assumption 4.1.

**Definition A.1** ($L$-Lipschitz)**.** The loss function $\mathcal{L}$ is $L$-Lipschitz in the parameter $\boldsymbol{w}$ if $\forall(\boldsymbol{x}, y) \in \mathcal{X} \times \mathcal{Y}$ and $\forall \boldsymbol{w}_1, \boldsymbol{w}_2 \in \mathcal{H}$,

$$|\mathcal{L}((\boldsymbol{x}, y), \boldsymbol{w}_1) - \mathcal{L}((\boldsymbol{x}, y), \boldsymbol{w}_2)| \leq L\|\boldsymbol{w}_1 - \boldsymbol{w}_2\|_2. \tag{7}$$

**Definition A.2** ($\alpha$-Strong Convexity)**.** The loss function $\mathcal{L}$ is $\alpha$-strong convex if $\forall(\boldsymbol{x}, y) \in \mathcal{X} \times \mathcal{Y}$ and $\forall \boldsymbol{w}_1, \boldsymbol{w}_2 \in \mathcal{H}$,

$$\mathcal{L}((\boldsymbol{x}, y), \boldsymbol{w}_1) \geq \mathcal{L}((\boldsymbol{x}, y), \boldsymbol{w}_2) + \langle \nabla\mathcal{L}((\boldsymbol{x}, y), \boldsymbol{w}_2), \boldsymbol{w}_1 - \boldsymbol{w}_2 \rangle + \frac{\alpha}{2}\|\boldsymbol{w}_1 - \boldsymbol{w}_2\|_2^2. \tag{8}$$

**Definition A.3** ($\beta$-Smoothness)**.** The loss function $\mathcal{L}$ is $\beta$-smooth if $\forall(\boldsymbol{x}, y) \in \mathcal{X} \times \mathcal{Y}$ and $\forall \boldsymbol{w}_1, \boldsymbol{w}_2 \in \mathcal{H}$,

$$\mathcal{L}((\boldsymbol{x}, y), \boldsymbol{w}_1) \leq \mathcal{L}((\boldsymbol{x}, y), \boldsymbol{w}_2) + \langle \nabla\mathcal{L}((\boldsymbol{x}, y), \boldsymbol{w}_2), \boldsymbol{w}_1 - \boldsymbol{w}_2 \rangle + \frac{\beta}{2}\|\boldsymbol{w}_1 - \boldsymbol{w}_2\|_2^2. \tag{9}$$

**Definition A.4** ($\gamma$-Hessian Lipschitz)**.** The loss function $\mathcal{L}$ is $\gamma$-Hessian Lipschitz in the parameter $\boldsymbol{w}$ if $\forall(\boldsymbol{x}, y) \in \mathcal{X} \times \mathcal{Y}$ and $\forall \boldsymbol{w}_1, \boldsymbol{w}_2 \in \mathcal{H}$,

$$|\nabla^2\mathcal{L}((\boldsymbol{x}, y), \boldsymbol{w}_1) - \nabla^2\mathcal{L}((\boldsymbol{x}, y), \boldsymbol{w}_2)| \leq \gamma\|\boldsymbol{w}_1 - \boldsymbol{w}_2\|_2. \tag{10}$$

# B. Relationship Between Certified Machine Unlearning and Differential Privacy

Differential privacy (Dwork, 2006) and certified unlearning share a common conceptual foundation. Both concepts aim to provide statistical indistinguishability. However, they focus on achieving indistinguishability in fundamentally different aspects of data handling and model behavior.

In differential privacy, the goal is to ensure that the information obtained from a randomized mechanism applied to a dataset is indistinguishable when a specific data sample is included versus when it is excluded. This statistical indistinguishability is achieved by bounding the influence of any single data point on the output of the mechanism. Formally, the randomized mechanism $\mathcal{M}$ satisfies $(\epsilon, \delta)$-differential privacy if, for any two neighboring datasets $\mathcal{D}$ and $\mathcal{D}'$ differing by at most one data point, and for any measurable set $\mathcal{T}$, the following holds.

**Definition B.1** (Differential Privacy)**.** A randomized mechanism $\mathcal{M}$ satisfies $(\epsilon, \delta)$-differential privacy if, for any two neighboring datasets $\mathcal{D}$ and $\mathcal{D}'$ differing in at most one data point, and for any measurable subset $\mathcal{T} \subseteq \mathcal{H}$:

$$\Pr\left(\mathcal{M}(\mathcal{D}) \in \mathcal{T}\right) \leq e^{\epsilon}\Pr\left(\mathcal{M}(\mathcal{D}') \in \mathcal{T}\right) + \delta,$$
$$\Pr\left(\mathcal{M}(\mathcal{D}') \in \mathcal{T}\right) \leq e^{\epsilon}\Pr\left(\mathcal{M}(D) \in \mathcal{T}\right) + \delta.$$

Thus, DP focuses on controlling the extent to which the output of the mechanism reveals information about any individual data sample.

In contrast, certified unlearning ensures statistical indistinguishability between the output of a retrained model and that of an unlearned model. The guarantee provided by certified unlearning, as defined in Definition 3.1, ensures that the behavior of the unlearned model closely approximates the retrained model within $(\epsilon, \delta)$ bounds.

Certified unlearning often employs DP-inspired techniques like the Gaussian mechanism to achieve guarantees. Specifically, certification requires finding an upper bound for the norm of the difference between a model retrained from scratch and one updated via the unlearning mechanism. This bound enables noise scaling according to the Gaussian mechanism ((Dwork, 2006), App. A) to satisfy $(\epsilon, \delta)$-certified unlearning, ensuring statistical indistinguishability between the unlearned and retrained models (Definition 3.1).

Previous works derive this upper bound based on the unlearning mechanism's algorithm. Many certified unlearning approaches rely on single-step second-order Newton updates under strong convexity assumptions (Guo et al., 2019; Sekhari et al., 2021; Zhang et al., 2024), where the Hessian is computed over the retain dataset $\mathcal{D}_r$ at the model trained on the full dataset $\boldsymbol{w}^*$.

### B.1. Certified Unlearning Using Newton Updates with Exact Data Samples

Certified unlearning methods often utilize statistical information, $\mathcal{S}(\cdot)$, such as the Hessian of the full dataset evaluated at $\boldsymbol{w}^*$ (Sekhari et al., 2021; Zhang et al., 2024). These methods randomize the unlearning mechanism by adding noise, following the Gaussian mechanism. The upper bound for the norm of the difference between the retrained and unlearned models is derived using retained data samples (Sekhari et al., 2021; Zhang et al., 2024).

Building on this framework, Sekhari et al. (Sekhari et al., 2021) derived an explicit upper bound for the norm difference, leveraging strong convexity and second-order information. This bound provides critical insights into the statistical guarantees of certified unlearning. In the following the upper bound between

**Lemma B.2** ((Sekhari et al., 2021) Lemma 3). *Let the loss function $\mathcal{L}$ satisfy Assumption 4.1. Suppose the training dataset $\mathcal{D}$ contains $n$ samples, and the forget dataset $\mathcal{D}_u \subseteq \mathcal{D}$ consists of $m$ samples. The norm of the difference between the model retrained from scratch, $\boldsymbol{w}_r^*$, and the model obtained using a second-order Newton update on the exact retain dataset $\mathcal{D}_r$, $\boldsymbol{w}_r$, is bounded above by*

$$\|\boldsymbol{w}_r^* - \boldsymbol{w}_r\|_2 \leq \frac{2\gamma L m^2}{\alpha^3 n^2}. \tag{11}$$

*Proof.* The proof can be found in the Supplementary Material of Sekhari et al. (Sekhari et al., 2021) under C.1. □

## C. Upper Bound for Norm of Difference Between Unlearning Updates (Proof of Theorem 4.2)

Let us focus on the Hessians of the loss function $\mathcal{L}$, calculated at the same model $\boldsymbol{w}$, for two different distributions, $\rho$ and $\nu$. The distributions $\rho$ and $\nu$ share the same support set, $\mathcal{X} \times \mathcal{Y}$. The Hessians corresponding to each distribution are defined as follows.

The Hessian of the loss function $\mathcal{L}$ under the distribution $\rho$, evaluated at the model $\boldsymbol{w}$, is given by

$$\mathbf{H}_\rho = \mathbb{E}_{(\boldsymbol{x},y)\sim\rho}\left[\nabla^2 \mathcal{L}((\boldsymbol{x},y),\boldsymbol{w})\right]. \tag{12}$$

Similarly, the Hessian of the loss function $\mathcal{L}$ under the distribution $\nu$, evaluated at the model $\boldsymbol{w}$, is given by

$$\mathbf{H}_\nu = \mathbb{E}_{(\boldsymbol{x},y)\sim\nu}\left[\nabla^2 \mathcal{L}((\boldsymbol{x},y),\boldsymbol{w})\right]. \tag{13}$$

Next, we focus on the spectral norm of the difference between these two Hessians. The following lemma provides an upper bound.

**Lemma C.1.** *If the loss function $\mathcal{L}$ satisfies the Assumption 4.1, then the following upper bound holds:*

$$\|\mathbf{H}_\rho - \mathbf{H}_\nu\|_2 \leq 2\beta \mathrm{TV}(\rho \| \nu) \tag{14}$$

*Proof.*

$$\|\mathbf{H}_\rho - \mathbf{H}_\nu\|_2 = \|\mathbb{E}_{(\boldsymbol{x},y)\sim\rho}\left[\nabla^2 \mathcal{L}((\boldsymbol{x},y),\boldsymbol{w})\right] - \mathbb{E}_{(\boldsymbol{x},y)\sim\nu}\left[\nabla^2 \mathcal{L}((\boldsymbol{x},y),\boldsymbol{w})\right]\|_2 \tag{15}$$

$$= \|\sum_{\boldsymbol{x}\in\mathcal{X}}\sum_{y\in\mathcal{Y}}\rho(\boldsymbol{x},y)\nabla^2 \mathcal{L}((\boldsymbol{x},y),\boldsymbol{w}) - \sum_{\boldsymbol{x}\in\mathcal{X}}\sum_{y\in\mathcal{Y}}\nu(\boldsymbol{x},y)\nabla^2 \mathcal{L}((\boldsymbol{x},y),\boldsymbol{w})\|_2 \tag{16}$$

$$= \|\sum_{\boldsymbol{x}\in\mathcal{X}}\sum_{y\in\mathcal{Y}}(\rho(\boldsymbol{x},y) - \nu(\boldsymbol{x},y))\nabla^2 \mathcal{L}((\boldsymbol{x},y),\boldsymbol{w})\|_2 \tag{17}$$

$$\leq \sum_{\boldsymbol{x}\in\mathcal{X}}\sum_{y\in\mathcal{Y}}|(\rho(\boldsymbol{x},y) - \nu(\boldsymbol{x},y))|\|\nabla^2 \mathcal{L}((\boldsymbol{x},y),\boldsymbol{w})\|_2 \tag{18}$$

$$\leq \beta \sum_{\boldsymbol{x}\in\mathcal{X}}\sum_{y\in\mathcal{Y}}|(\rho(\boldsymbol{x},y) - \nu(\boldsymbol{x},y))| \tag{19}$$

$$= 2\beta \mathrm{TV}(\rho \| \nu) \tag{20}$$

In the above, Equation (18) follows from the sub-multiplicativity and sub-additivity of the matrix norm. Equation (19) holds due to the $\beta$-smoothness property of the loss function $\mathcal{L}$, and Equation (20) follows from the definition of the Total Variation distance. $\qquad\square$

Building on the discussion of the Hessians for the distributions $\rho$ and $\nu$, we now turn our attention to their empirical counterparts. The empirical Hessian matrices can be represented as follows. Let the training dataset $\mathcal{D}$ consist of $n_1$ samples drawn from the distribution $\rho$, and let $\mathcal{D}_s$ consist of $n_2$ samples drawn from the distribution $\nu$. Then, if $n_1$ and $n_2$ are sufficiently large, we can make the following statement based on the law of large numbers.

$$\mathbf{H}_{\mathcal{D}} = \frac{1}{n_1} \sum_{(\boldsymbol{x},y)\in\mathcal{D}} \nabla^2 \mathcal{L}((\boldsymbol{x},y),\boldsymbol{w}) \approx \mathbb{E}_{(\boldsymbol{x},y)\sim\rho}\left[\nabla^2\mathcal{L}((\boldsymbol{x},y),\boldsymbol{w})\right], \tag{21}$$

$$\mathbf{H}_{\mathcal{D}_s} = \frac{1}{n_2} \sum_{(\boldsymbol{x},y)\in\mathcal{D}_s} \nabla^2 \mathcal{L}((\boldsymbol{x},y),\boldsymbol{w}) \approx \mathbb{E}_{(\boldsymbol{x},y)\sim\nu}\left[\nabla^2\mathcal{L}((\boldsymbol{x},y),\boldsymbol{w})\right]. \tag{22}$$

After establishing the empirical Hessian matrices and their dependence on datasets $\mathcal{D}$ and $\mathcal{D}_s$, we now state the following result, which provides a bound on the spectral norm of their scaled difference. This result directly follows from the assumptions on the loss function $\mathcal{L}$ and the distributions $\rho$ and $\nu$.

**Lemma C.2.** *Suppose the loss function $\mathcal{L}$ satisfies Assumption 4.1. Let the training dataset $\mathcal{D}$ consist of $n_1$ samples drawn from the distribution $\rho$, and let the surrogate dataset $\mathcal{D}_s$ consist of $n_2$ samples drawn from the distribution $\nu$. Assuming that $n_1$ and $n_2$ are sufficiently large, the following bound holds.*

$$\|n_1\mathbf{H}_{\mathcal{D}} - n_2\mathbf{H}_{\mathcal{D}_s}\|_2 \le (n_1 - n_2)\beta + 2n_2\beta\mathrm{TV}(\rho \,\|\, \nu) \tag{23}$$

*Proof.* Without loss of generality assume that $n_1 \ge n_2$

$$\|n_1\mathbf{H}_{\mathcal{D}} - n_2\mathbf{H}_{\mathcal{D}_s}\|_2 = \|(n_1 - n_2)\mathbf{H}_{\mathcal{D}} + n_2\mathbf{H}_{\mathcal{D}} - n_2\mathbf{H}_{\mathcal{D}_s}\|_2 \tag{24}$$

$$\le (n_1 - n_2)\|\mathbf{H}_{\mathcal{D}}\|_2 + n_2\|\mathbf{H}_{\mathcal{D}} - \mathbf{H}_{\mathcal{D}_s}\|_2 \tag{25}$$

$$\approx (n_1 - n_2)\|\mathbf{H}_{\mathcal{D}}\|_2 + n_2\|\mathbf{H}_{\rho} - \mathbf{H}_{\nu}\|_2 \tag{26}$$

$$\le (n_1 - n_2)\beta + 2n_2\beta\mathrm{TV}(\rho \,\|\, \nu) \tag{27}$$

Here, the first inequality uses the triangle inequality for matrix norms. The approximation in the third step relies on the assumption that the empirical Hessian matrices $\mathbf{H}_{\mathcal{D}}$ and $\mathbf{H}_{\mathcal{D}_s}$ converge to their population counterparts $\mathbf{H}_{\rho}$ and $\mathbf{H}_{\nu}$, respectively, when $n_1$ and $n_2$ are sufficiently large by the law of large numbers. The last inequality holds because of the Lemma C.1. $\qquad\square$

To proceed with the main proof, we introduce the following lemma as a key tool. This lemma provides an upper bound on the spectral norm of the inverse of the weighted difference of Hessians.

**Lemma C.3.** *Suppose the loss function $\mathcal{L}$ satisfies Assumption 4.1. Let the training dataset $\mathcal{D}$ consist of $n$ samples, and the forget dataset $\mathcal{D}_u$ consist of $m$ samples. If $n > \frac{m\beta}{\alpha}$, then the following bound holds:*

$$\|(n\mathbf{H}_{\mathcal{D}} - m\mathbf{H}_{\mathcal{D}_u})^{-1}\|_2 \le \frac{1}{n\alpha - m\beta} \tag{28}$$

*Proof.* By using the reverse triangle inequality we know that,

$$\|(n\mathbf{H}_{\mathcal{D}} - m\mathbf{H}_{\mathcal{D}_u})\|_2 \ge |n\|\mathbf{H}_{\mathcal{D}}\|_2 - m\|\mathbf{H}_{\mathcal{D}_u}\|_2|. \tag{29}$$

If $n > \frac{m\beta}{\alpha}$ then the inner term will be positive because the spectral norms of Hessians are between $\alpha$ and $\beta$ by the Assumption 4.1 (strong convexity and smoothness). Therefore,

$$|n\|\mathbf{H}_{\mathcal{D}}\|_2 - m\|\mathbf{H}_{\mathcal{D}_u}\|_2| = n\|\mathbf{H}_{\mathcal{D}}\|_2 - m\|\mathbf{H}_{\mathcal{D}_u}\|_2 \tag{30}$$

$$\geq n\alpha - m\beta. \tag{31}$$

Having this lower bound on the spectral norm of $\|(n\mathbf{H}_{\mathcal{D}} - m\mathbf{H}_{\mathcal{D}_u})\|_2$, we can conclude on the following upper bound.

$$\|(n\mathbf{H}_{\mathcal{D}} - m\mathbf{H}_{\mathcal{D}_u})^{-1}\|_2 \leq \frac{1}{n\alpha - m\beta} \tag{32}$$

$\square$

In this section, we focus on quantifying the difference between the models approximated using the exact dataset and the surrogate dataset. By leveraging the previously introduced tools, we can establish an upper bound on the norm of the difference between these two models.

**Lemma C.4.** *Suppose the loss function $\mathcal{L}$ satisfies Assumption 4.1. Let $\boldsymbol{w}_r$ denote the model approximated using the exact dataset $\mathcal{D}$ of size $n_1$ and the forget set $\mathcal{D}_u$ of size $m$. Similarly, let $\widehat{\boldsymbol{w}}_r$ denote the model approximated using the surrogate dataset $\mathcal{D}_s$ of size $n_2$. Also, assume that the $n_1$ and $n_2$ are sufficiently large and $n_i \geq \frac{m\beta}{\alpha}$ where $i \in \{1, 2\}$ . Then, the norm of the difference between the approximated models is upper bounded as follows:*

$$\|\boldsymbol{w}_r - \widehat{\boldsymbol{w}}_r\|_2 \leq \frac{m(n_1 - n_2)\beta + 2mn_2\beta\mathrm{TV}(\rho \,\|\, \nu)}{(n_1\alpha - m\beta)(n_2\alpha - m\beta)}\|\nabla\mathcal{L}(\mathcal{D}_u, \boldsymbol{w}^*)\|_2 \tag{33}$$

*Proof.* Let's start from the applied update to the trained model $\boldsymbol{w}^*$ having the exact training data samples and the forget set.

$$\boldsymbol{w}_r = \boldsymbol{w}^* + \frac{m}{n_1 - m}\left(\frac{n_1\mathbf{H}_{\mathcal{D}} - m\mathbf{H}_{\mathcal{D}_u}}{n_1 - m}\right)^{-1}\nabla\mathcal{L}(\mathcal{D}_u, \boldsymbol{w}^*) \tag{34}$$

$$= \boldsymbol{w}^* + m\left(n_1\mathbf{H}_{\mathcal{D}} - m\mathbf{H}_{\mathcal{D}_u}\right)^{-1}\nabla\mathcal{L}(\mathcal{D}_u, \boldsymbol{w}^*) \tag{35}$$

The model achieved after applying the update with the surrogate dataset $\mathcal{D}_s$ is

$$\widehat{\boldsymbol{w}}_r = \boldsymbol{w}^* + \frac{m}{n_2 - m}\left(\frac{n_2\mathbf{H}_{\mathcal{D}_s} - m\mathbf{H}_{\mathcal{D}_u}}{n_2 - m}\right)^{-1}\nabla\mathcal{L}(\mathcal{D}_u, \boldsymbol{w}^*) \tag{36}$$

$$= \boldsymbol{w}^* + m\left(n_2\mathbf{H}_{\mathcal{D}_s} - m\mathbf{H}_{\mathcal{D}_u}\right)^{-1}\nabla\mathcal{L}(\mathcal{D}_u, \boldsymbol{w}^*) \tag{37}$$

$$\|\boldsymbol{w}_r - \widehat{\boldsymbol{w}}_r\|_2 = \|m\left((n_1\mathbf{H}_{\mathcal{D}} - m\mathbf{H}_{\mathcal{D}_u})^{-1} - (n_2\mathbf{H}_{\mathcal{D}_s} - m\mathbf{H}_{\mathcal{D}_u})^{-1}\right)\nabla\mathcal{L}(\mathcal{D}_u, \boldsymbol{w}^*)\|_2 \tag{38}$$

$$\leq m\left\|\left((n_1\mathbf{H}_{\mathcal{D}} - m\mathbf{H}_{\mathcal{D}_u})^{-1} - (n_2\mathbf{H}_{\mathcal{D}_s} - m\mathbf{H}_{\mathcal{D}_u})^{-1}\right)\right\|_2\|\nabla\mathcal{L}(\mathcal{D}_u, \boldsymbol{w}^*)\|_2 \tag{39}$$

$$\leq m\left\|(n_2\mathbf{H}_{\mathcal{D}_s} - m\mathbf{H}_{\mathcal{D}_u})^{-1}\right\|_2\|(n_1\mathbf{H}_{\mathcal{D}} - m\mathbf{H}_{\mathcal{D}_u}) - (n_2\mathbf{H}_{\mathcal{D}_s} - m\mathbf{H}_{\mathcal{D}_u})\|_2$$
$$\left\|(n_1\mathbf{H}_{\mathcal{D}} - m\mathbf{H}_{\mathcal{D}_u})^{-1}\right\|_2\|\nabla\mathcal{L}(\mathcal{D}_u, \boldsymbol{w}^*)\|_2 \tag{40}$$

$$= m\left\|(n_2\mathbf{H}_{\mathcal{D}_s} - m\mathbf{H}_{\mathcal{D}_u})^{-1}\right\|_2\|(n_1\mathbf{H}_{\mathcal{D}} - n_2\mathbf{H}_{\mathcal{D}_s})\|_2$$
$$\left\|(n_1\mathbf{H}_{\mathcal{D}} - m\mathbf{H}_{\mathcal{D}_u})^{-1}\right\|_2\|\nabla\mathcal{L}(\mathcal{D}_u, \boldsymbol{w}^*)\|_2 \tag{41}$$

$$\leq \frac{m(n_1 - n_2)\beta + 2mn_2\beta\mathrm{TV}(\rho \,\|\, \nu)}{(n_1\alpha - m\beta)(n_2\alpha - m\beta)}\|\nabla\mathcal{L}(\mathcal{D}_u, \boldsymbol{w}^*)\|_2 \tag{42}$$

The last inequality holds by using Lemma C.2 and Lemma C.3. $\square$

With all the necessary tools established, we are now ready to prove the main result, Theorem 4.2. This theorem quantifies the relationship between the retrained model over the retained samples and the model approximated using the surrogate dataset, providing an upper bound on their difference.

**Theorem C.5** (Proof of Theorem 4.2). *Consider a loss function $\mathcal{L}$ satisfying Assumption 4.1, and a surrogate dataset $\mathcal{D}_s$ with $n_2$ samples drawn from a distribution $\nu$, to mimic the true dataset $\mathcal{D}$ with $n_1$ drawn from a distribution $\rho$, over the support set $\mathcal{X} \times \mathcal{Y}$. Define the retrained model over the retained samples as $\boldsymbol{w}_r^*$ and the model achieved after unlearning as $\widehat{\boldsymbol{w}}_r$. Also, assume that the $n_1$ and $n_2$ are sufficiently large and $n_i \geq \frac{m\beta}{\alpha}$ where $i \in \{1, 2\}$. Then, the following upper bound holds,*

$$\|\boldsymbol{w}_r^* - \widehat{\boldsymbol{w}}_r\|_2 \leq \frac{2\gamma L m^2}{\alpha^3 n_1^2} + \frac{m(n_1 - n_2)\beta + 2mn_2\beta \mathrm{TV}(\rho \parallel \nu)}{(n_1\alpha - m\beta)(n_2\alpha - m\beta)}\|\nabla\mathcal{L}(\mathcal{D}_u, \boldsymbol{w}^*)\|_2 \tag{43}$$

*Proof.* By the triangle inequality,

$$\|\boldsymbol{w}_r^* - \widehat{\boldsymbol{w}}_r\|_2 = \|\boldsymbol{w}_r^* - \boldsymbol{w}_r + \boldsymbol{w}_r - \widehat{\boldsymbol{w}}_r\|_2 \tag{44}$$
$$\leq \|\boldsymbol{w}_r^* - \boldsymbol{w}_r\|_2 + \|\boldsymbol{w}_r - \widehat{\boldsymbol{w}}_r\|_2 \tag{45}$$

Then by utilizing the Lemma C.4 and Lemma B.2 the upper bound is proven. $\square$

### C.1. Proof of Theorem 4.3

**Theorem C.6** (Proof of Theorem 4.3). *Consider a dataset $\mathcal{D}$ where data samples follow the distribution $\rho$, and a surrogate dataset $\mathcal{D}_s$ where data samples follow the distribution $\nu$. Given a forget set $\mathcal{D}_u \subseteq \mathcal{D}$, and the hypothesis set $\mathcal{H}$, the unlearning mechanism $\widehat{\mathcal{U}}$ (Algorithm 1) satisfies $(\epsilon, \delta)$-certified unlearning. For any $\mathcal{T} \subseteq \mathcal{H}$,*

$$\Pr\left(\widehat{\mathcal{U}}(\mathcal{D}_u, \mathcal{A}(\mathcal{D}), \mathcal{S}(\mathcal{D}_s)) \in \mathcal{T}\right) \leq e^\epsilon \Pr\left(\widehat{\mathcal{U}}(\emptyset, \mathcal{A}(\mathcal{D}_r), \mathcal{S}(\mathcal{D}_r)) \in \mathcal{T}\right) + \delta,$$
$$\Pr\left(\widehat{\mathcal{U}}(\emptyset, \mathcal{A}(\mathcal{D}_r), \mathcal{S}(\mathcal{D}_r)) \in \mathcal{T}\right) \leq e^\epsilon \Pr\left(\widehat{\mathcal{U}}(\mathcal{D}_u, \mathcal{A}(\mathcal{D}), \mathcal{S}(\mathcal{D}_s)) \in \mathcal{T}\right) + \delta. \tag{46}$$

*Proof.* Let $\boldsymbol{w}^* = \mathcal{A}(\mathcal{D})$ denote the model trained on the whole train dataset $\mathcal{D}$ with $n_1$ number of samples following the distribution $\rho$, $\boldsymbol{w}_r^* = \mathcal{A}(\mathcal{D}_r)$ the model retrained from scratch over the retain dataset $\mathcal{D}_r = \mathcal{D} \backslash \mathcal{D}_u$ where $\mathcal{D}_u$ is the forget set with $m$ number of samples, and $\widehat{\boldsymbol{w}}_r$ the model approximated retrained model after the single step second-order Newton update utilizing the surrogate dataset $\mathcal{D}_s$ with $n_2$ number of samples following distribution $\nu$. Assume that the loss function used is satisfying the Assumption 4.1, $n_1$ and $n_2$ are sufficiently large and $n_i \geq \frac{m\beta}{\alpha}$ where $i \in \{1, 2\}$. Also the support sets of distributions are the same $\mathcal{X} \times \mathcal{Y}$.

The $\widehat{\boldsymbol{w}}_r$ defined as

$$\widehat{\boldsymbol{w}}_r = \boldsymbol{w}^* + \frac{m}{n_2 - m}\left(\frac{n_2\mathbf{H}_{\mathcal{D}_s} - m\mathbf{H}_{\mathcal{D}_u}}{n_2 - m}\right)^{-1}\nabla\mathcal{L}(\mathcal{D}_u, \boldsymbol{w}^*). \tag{47}$$

By applying Theorem C.5 we can observe the following upper bound,

$$\|\boldsymbol{w}_r^* - \widehat{\boldsymbol{w}}_r\|_2 \leq \frac{2\gamma L m^2}{\alpha^3 n_1^2} + \frac{m(n_1 - n_2)\beta + 2mn_2\beta \mathrm{TV}(\rho \parallel \nu)}{(n_1\alpha - m\beta)(n_2\alpha - m\beta)}\|\nabla\mathcal{L}(\mathcal{D}_u, \boldsymbol{w}^*)\|_2 = \Delta. \tag{48}$$

The Algorithm 1 introduces the Gaussian noise to achieve indistinguishability between the model retrained from scratch $\boldsymbol{w}_r^*$ and the model achieved after the unlearning $\widehat{\boldsymbol{w}}_r$. Let's define $(\boldsymbol{w}_r^*)' = \widehat{\mathcal{U}}(\emptyset, \mathcal{A}(\mathcal{D}_r), \mathcal{S}(\mathcal{D}_r)) = \boldsymbol{w}_r^* + \boldsymbol{n}$ and $\widehat{\boldsymbol{w}}_r' = \widehat{\mathcal{U}}(\mathcal{D}_u, \mathcal{A}(\mathcal{D}), \mathcal{S}(\mathcal{D}_s)) = \widehat{\boldsymbol{w}}_r + \boldsymbol{n}$ where the Gaussian noise $\boldsymbol{n} \sim \mathcal{N}(\boldsymbol{0}, \sigma^2\mathbf{I})$ with $\sigma = (\Delta/\epsilon)\sqrt{2\log(1.25/\delta)}$. Following the proof from Dwork et al. ((Dwork, 2006) Theorem A.1), we can prove that for any set $\mathcal{T} \subseteq \mathcal{H}$,

$$\Pr\left(\widehat{\boldsymbol{w}}_r' \in \mathcal{T}\right) \leq e^\epsilon \Pr\left((\boldsymbol{w}_r^*)' \in \mathcal{T}\right) + \delta,$$
$$\Pr\left((\boldsymbol{w}_r^*)' \in \mathcal{T}\right) \leq e^\epsilon \Pr\left(\widehat{\boldsymbol{w}}_r' \in \mathcal{T}\right) + \delta. \tag{49}$$

$\square$

# D. Approximating Kullbeck-Leiber Distance

## D.1. Proof of Corollary 4.4

**Corollary D.1** (Proof of Corollary 4.4). *Under the same assumptions and definitions in Theorem 4.2, the following upper bound holds:*

$$
\|\boldsymbol{w}_r^* - \widehat{\boldsymbol{w}}_r\|_2 \leq \frac{2\gamma L^2 m^2}{\alpha^3 n^2} + \frac{m(n_1 - n_2)\beta\|\nabla\mathcal{L}(\mathcal{D}_u, \boldsymbol{w}^*)\|_2}{(n_1\alpha - m\beta)(n_2\alpha - m\beta)}
$$
$$
+ \left( \frac{2mn_2\beta\sqrt{1 - \exp(-\text{KL}(\nu \parallel \rho))}}{(n_1\alpha - m\beta)(n_2\alpha - m\beta)} \cdot \|\nabla\mathcal{L}(\mathcal{D}_u, \boldsymbol{w}^*)\|_2 \right)
$$
$$
= \Delta
$$

(50)

*Proof.* The total variation distance is symmetric therefore,

$$
\text{TV}(\rho \parallel \nu) = \text{TV}(\nu \parallel \rho)
$$

(51)

Also, by the Bretagnolle-Huber Inequality (Bretagnolle & Huber, 1979),

$$
\text{TV}(\nu \parallel \rho) \leq \sqrt{1 - e^{-\text{KL}(\nu \parallel \rho)}}
$$

(52)

By replacing the total variation distance with this upper bound we prove the Corollary 4.4. $\square$

We need this upper bound utilizing the KL divergence between the surrogate and the exact data distributions because we have the surrogate samples. By having the surrogate samples we can approximate the expected values calculated over surrogate samples utilizing Monte-Carlo approximation.

## D.2. Derivations for Proposition 4.5

**Proposition D.2** (Derivation of Proposition 4.5). *Let $f(\boldsymbol{w}, \boldsymbol{x})$ output a probability simplex over classes for a data sample $\boldsymbol{x}$, parameterized by $\boldsymbol{w}$. Given trained models $\boldsymbol{w}^*$ and $\tilde{\boldsymbol{w}}^*$, where $\boldsymbol{w}^*$ is trained on $\mathcal{D}$ and $\tilde{\boldsymbol{w}}^*$ on $\mathcal{D}_s$. Also, data samples from $\mathcal{D}$ and $\mathcal{D}_s$ follow distributions $\rho$ and $\nu$, respectively. the KL divergence $\text{KL}(\nu \parallel \rho)$ can be decomposed as,*

$$
\text{KL}(\nu \parallel \rho) \approx \frac{1}{n} \sum_{(\boldsymbol{x},y)\in\mathcal{D}_s} \log \frac{f(\tilde{\boldsymbol{w}}^*, \boldsymbol{x})_y}{f(\boldsymbol{w}^*, \boldsymbol{x})_y} + \text{KL}(\nu(\boldsymbol{x}) \parallel \rho(\boldsymbol{x}))
$$

(53)

*Proof.* Starting with the definition,

$$
\text{KL}(\nu \parallel \rho) = \sum_{(\boldsymbol{x},y)\in\mathcal{X}\times\mathcal{Y}} \nu(\boldsymbol{x}, y) \log \frac{\nu(y|\boldsymbol{x})}{\rho(y|\boldsymbol{x})} + \sum_{(\boldsymbol{x})\in\mathcal{X}} \nu(\boldsymbol{x}) \log \frac{\nu(\boldsymbol{x})}{\rho(\boldsymbol{x})}
$$

(54)

The conditional probabilities can be approximated by classifiers. Let's say the classifier model trained on the exact dataset (representing conditional distribution for exact distribution) is $\boldsymbol{w}^*$. This model is already given to us for the unlearning. To represent the nominator we need to have an another model trained on the surrogate dataset provided, let's say after the training on the surrogate dataset we achieve the model $\tilde{\boldsymbol{w}}^*$. Then we can approximate (54) as

$$
\text{KL}(\nu \parallel \rho) \approx \sum_{(\boldsymbol{x},y)\in\mathcal{X}\times\mathcal{Y}} \nu(\boldsymbol{x}, y) \log \frac{f(\tilde{\boldsymbol{w}}^*, \boldsymbol{x})_y}{f(\boldsymbol{w}^*, \boldsymbol{x})_y} + \text{KL}(\nu(\boldsymbol{x}) \parallel \rho(\boldsymbol{x}))
$$

(55)

By using Monte-Carlo approximation because we have access to the surrogate data samples we can further approximate (55) and prove the derivation.

$$
\text{KL}(\nu \parallel \rho) \approx \frac{1}{n} \sum_{(\boldsymbol{x},y)\in\mathcal{D}_s} \log \frac{f(\tilde{\boldsymbol{w}}^*, \boldsymbol{x})_y}{f(\boldsymbol{w}^*, \boldsymbol{x})_y} + \text{KL}(\nu(\boldsymbol{x}) \parallel \rho(\boldsymbol{x}))
$$

(56)

$\square$

**D.3. Energy Based Modeling, Stochastic Gradient Langevin Dynamics and Proposition 4.6**

Without direct access to exact samples from the target distribution, we approximate the input marginal KL divergence using energy-based modeling, as introduced in (Grathwohl et al., 2019). Energy-based models (EBMs) provide a flexible framework for modeling complex distributions by associating an energy score with each input, which corresponds to the unnormalized log-probability of the input under the target distribution. The normalized target distribution, denoted as $\rho(\boldsymbol{x})$, can be expressed as:

$$\rho(\boldsymbol{x}) = \frac{\exp(-E(\boldsymbol{x}))}{Z} \tag{57}$$

Here, $E(\boldsymbol{x})$ represents the energy function, which is defined as follows,

$$E(\boldsymbol{x}) = -\log \sum_{y \in \mathcal{Y}} \exp(f(\boldsymbol{w}^*, \boldsymbol{x})_y) \tag{58}$$

where $f(\boldsymbol{w}^*, \boldsymbol{x})_y$ corresponds to the logit score (or unnormalized log-probability) for the label $y$ given the input $\boldsymbol{x}$ under the model $\boldsymbol{w}^*$. The summation is taken over the support of the output label space $\mathcal{Y}$. The normalization constant $Z = \int \exp(-E(\boldsymbol{x})) \, d\boldsymbol{x}$ ensures that $\rho(\boldsymbol{x})$ is a valid probability distribution. However, evaluating $Z$ is computationally intractable due to the high-dimensional integral over the input space.

To circumvent the intractability of computing $Z$, we employ Stochastic Gradient Langevin Dynamics (SGLD), a popular sampling technique for EBMs. SGLD enables approximate sampling from $\rho(\boldsymbol{x})$ by iteratively updating samples based on the gradient of the energy function. Specifically, the update rule for the samples $\boldsymbol{x}$ at step $i$ is given by:

$$\boldsymbol{x}_{i+1} = \boldsymbol{x}_i - \frac{\mu}{2} \frac{\partial E(\boldsymbol{x})}{\partial \boldsymbol{x}} + \varepsilon \tag{59}$$

where $\varepsilon \sim \mathcal{N}(0, \mu)$ is Gaussian noise, and $\mu$ represents the step size or learning rate. The negative gradient term $-\frac{\partial E(\boldsymbol{x})}{\partial \boldsymbol{x}}$ directs the samples toward regions of lower energy (higher likelihood), while the Gaussian noise ensures exploration of the input space to avoid convergence to local minima. The process begins with initializing $\boldsymbol{x}_0$ from a prior distribution over the input space, which is often chosen to be uniform for simplicity and generality.

By iteratively applying this update rule, the generated samples approximate the target distribution $\rho(\boldsymbol{x})$ without requiring explicit computation of the normalization constant $Z$. Given the collected samples from the distribution $\rho(\boldsymbol{x})$ along with the surrogate data samples, we then approximate the input marginal KL divergence using a Donsker-Varadhan variational representation (Donsker & Varadhan, 1983) of KL divergence.

# E. Parameter Study and Implementation

We systematically evaluate our approach on both synthetic and real-world datasets to demonstrate its effectiveness in achieving certified unlearning. Unless stated otherwise, we use a linear training model with privacy parameters $\epsilon = 5e^3$ and $\delta = 1$, a forget ratio of 0.1, and an $L_2$ regularization constant of $\lambda = 0.01$. The loss function is assumed to be $\alpha$-strongly convex, $L$-Lipschitz, $\beta$-smooth, and $\gamma$-Hessian Lipschitz. Following prior works (Koh & Liang, 2017; Wu et al., 2023b;a; Zhang et al., 2024), we set $\alpha$, $L$, $\beta$, and $\gamma$ for each experimental setting as hyperparameters. We set $\alpha = 1 + \lambda$, $L = 1$, $\beta = 1$ and $\gamma = 1$. Even if the added noise does not follows the exact theoretical requirements, it does not affect the theoretical soundness of the paper.

For the sampling from the marginal distribution of the exact data, we used Stochastic Gradient Langevin Dynamics (SGLD) with step size 0.02 and generate 1000 samples. For each sample random update is applied 4000 iteration for each generated sample.

After sampling done to estimate the KL divergence via Donsker Varadhan variational bound, we trained a a network with three linear layers for a 500 epochs with learning rate 0.0001 using Adam optimizer.

# F. Additional Synthetic and Real-World Dataset Experiments

We conducted experiments on both synthetic and real datasets to justify the heuristic KL-divergence estimation (Section 4.2) and the corresponding empirical unlearning error $\hat{\Delta}$. In Figures 3 and 4, we plot both the "exact" and "approximated"

results (our heuristic method) for the KL-divergence and the respective noise $\sigma$. For the synthetic data experiments, the exact KL is computed using its closed-form for Gaussians. For real data experiments, since the exact KL divergence was not available, we estimated it using the Donsker-Varadhan bound as a reference, leveraging both exact and surrogate data samples. Figure 3, Figure 4, Tables 8 and 9 show our approximations closely match exact values.

| $\zeta$ | Method | Train | Test | Retain | Forget | MIA | RT | $\Delta$ | $\hat{\Delta}$ |
|---|---|---|---|---|---|---|---|---|---|
| – | Original | 78.2±0.2% | 74.0±0.3% | 78.2±0.2% | 78.6±0.4% | 47.6±0.2% | – | – | – |
| – | Retrain | 77.2±0.3% | 71.8±0.2% | 77.7±0.5% | 73.2±0.4% | 47.4±0.2% | 10±1 | – | – |
| – | Unlearn (+) | 77.4±0.6% | 72.1±0.7% | 76.9±0.8% | 74.1±0.3% | 47.5±0.4% | 10±2 | 0.02 | – |
| 0.02 | **Unlearn (-)** | **77.5±0.2%** | **72.3±0.5%** | **77.8±0.1%** | **74.5±0.2%** | **49.1±0.4%** | **9±2** | **0.23** | **0.21±0.12** |
| 0.04 | **Unlearn (-)** | **77.4±0.6%** | **72.4±0.3%** | **77.2±0.4%** | **74.3±0.3%** | **49.2±0.2%** | **10±3** | **0.31** | **0.35±0.23** |
| 0.06 | **Unlearn (-)** | **77.4±0.5%** | **72.4±0.6%** | **77.5±0.2%** | **74.2±0.5%** | **48.7±0.7%** | **11±2** | **0.37** | **0.38±0.13** |
| 0.08 | **Unlearn (-)** | **77.5±0.7%** | **72.3±0.2%** | **77.4±0.7%** | **74.2±0.3%** | **48.1±0.5%** | **9±1** | **0.4** | **0.39±0.15** |
| 0.1 | **Unlearn (-)** | **77.4±0.4%** | **72.3±0.1%** | **77.7±0.4%** | **74.1±0.9%** | **48.2±0.2%** | **12±3** | **0.41** | **0.41±0.08** |

*Table 8.* Evaluation of unlearning performance while varying the off-diagonal elements ($\zeta$) of the unit covariance on synthetic dataset.

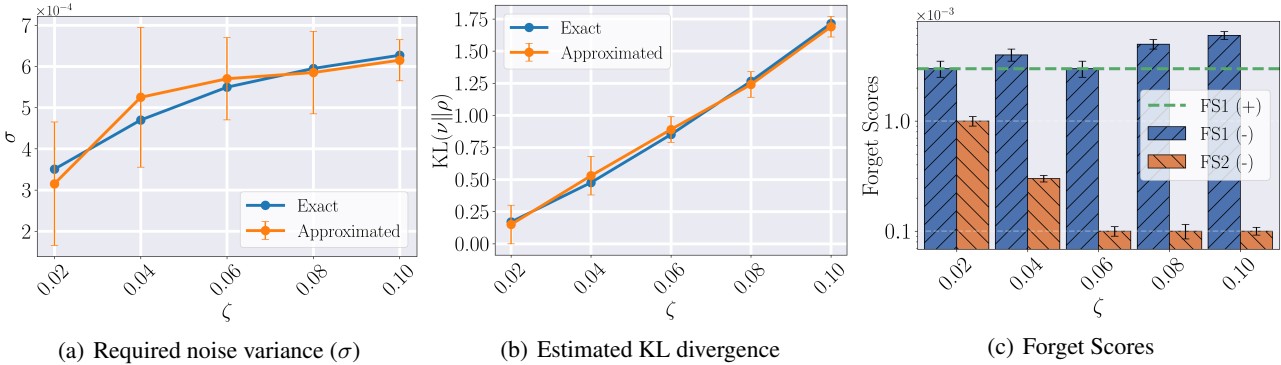

(a) Required noise variance ($\sigma$)  (b) Estimated KL divergence  (c) Forget Scores

*Figure 3.* **(a)** Required noise variance $\sigma$ for certified unlearning on synthetic data as a function of the off-diagonal elements of the covariance matrix ($\zeta$). Both exact and heuristic (approximate) estimates are shown based on KL divergence. **(b)** Estimated KL divergence vs. $\zeta$. Exact values use the closed-form Gaussian KL divergence, approximate values use our heuristic based on model parameters and surrogate data. **(c)** Forget scores achieved for synthetic datasets for varying off-diagonal elements of the covariance matrix ($\zeta$).

| $\xi$ | Method | Train | Test | Retain | Forget | MIA | RT | $\Delta$ | $\hat{\Delta}$ |
|---|---|---|---|---|---|---|---|---|---|
| 13 | Original | 85.9±0.8% | 73.3±0.1% | 85.4±0.5% | 84.6±0.2% | – | – | – | – |
| | Retrain | 86.7±0.2% | 73.7±0.8% | 87.7±0.5% | 75.6±0.2% | 51.5±0.7% | 70±1 | – | – |
| | Unlearn (+) | 84.1±0.9% | 71.7±0.4% | 85.2±0.6% | 73.2±0.7% | 51.4±0.4% | 15±3 | 0.02 | – |
| | **Unlearn (-)** | **84.7±0.4%** | **72.2±0.9%** | **85.3±0.7%** | **73.6±0.5%** | **51.8±0.3%** | **15±2** | **0.49±0.03** | **0.51±0.04** |
| 36 | Original | 85.5±0.1% | 76.2±0.7% | 85.6±0.8% | 83.1±0.3% | – | – | – | – |
| | Retrain | 84.1±0.4% | 75.1±0.8% | 86.1±0.9% | 71.5±0.5% | 50.3±0.1% | 20±1 | – | – |
| | Unlearn (+) | 84.9±0.1% | 75.6±0.1% | 85.1±0.1% | 71.5±0.3% | 50.8±0.5% | 18±1 | 0.02 | – |
| | **Unlearn (-)** | **84.9±0.8%** | **75.3±0.9%** | **85.1±0.4%** | **72.4±0.8%** | **50.2±0.8%** | **19±3** | **0.43±0.02** | **0.41±0.02** |
| 100 | Original | 84.7±0.5% | 71.3±0.3% | 84.9±0.4% | 83.6±0.5% | – | – | – | – |
| | Retrain | 82.9±0.3% | 76.0±0.2% | 84.2±0.8% | 71.1±0.5% | 52.5±0.4% | 19±2 | – | – |
| | Unlearn (+) | 83.8±0.5% | 75.7±0.7% | 85.1±0.1% | 72.2±0.8% | 52.0±0.4% | 21±1 | 0.02 | – |
| | **Unlearn (-)** | **83.7±0.4%** | **75.6±0.3%** | **85.0±0.7%** | **72.0±0.6%** | **51.9±0.2%** | **16±3** | **0.25±0.02** | **0.31±0.07** |

*Table 9.* Evaluation of unlearning performance while varying Dirichlet parameters ($\xi$) on StanfordDogs dataset.

To address whether the approximate certificate is practically useful, we report both $\Delta$ (exact) and $\hat{\Delta}$. The results confirm that even if the estimated bounds grow, model performance aligns with Unlearn(+). For completeness, we ran multiple trials with different seeds, included error bars in all figures, and included error margins into the tables demonstrating consistency across repeated experiments. Overall, these findings validate the heuristic's reliability and practical utility in estimating KL divergence and unlearning error.

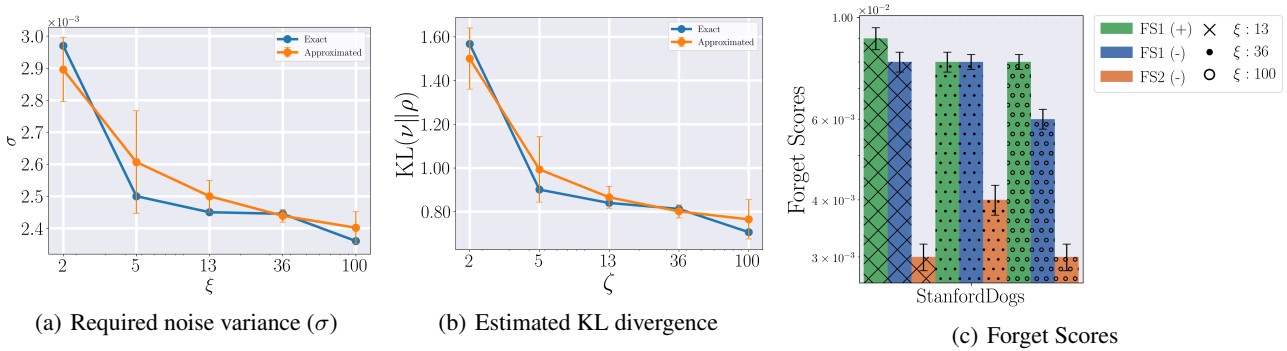

(a) Required noise variance ($\sigma$)  (b) Estimated KL divergence  (c) Forget Scores

*Figure 4.* **(a)** Required noise variance $\sigma$ for certified unlearning on StanfordDogs as a function of the Dirichlet parameter ($\xi$). Both exact and heuristic (approximate) estimates are shown based on KL divergence. **(b)** Estimated KL divergence vs. $\xi$. Exact values are calculated by using the exact and surrogate data samples, approximate values use our heuristic based on model parameters and surrogate data. **(c)** Forget scores achieved for StanfordDogs dataset for varying Dirichlet parameter ($\xi$).

