# OpenReview forum: "A Certified Unlearning Approach without Access to Source Data"
_ICML.cc/2025/Conference — ICML 2025 poster_

### Official Review · Reviewer_B4Hr · 2025-02-22

**Overall Recommendation:** 2

**Summary:**

This work focuses on certified unlearning when source data are unavailable due to resource limitation or regulation constraints. Instead, this work uses surrogate datasets to guide unlearning process, where the dataset mimics the original set to a certain extent. The indistinguishability guarantee is based on distance between those two datasets, and works for classification problem.

**Claims And Evidence:**

Yes.

**Essential References Not Discussed:**

To my best knowledge, authors covered most essential works related to this topic.

**Experimental Designs Or Analyses:**

The datasets (e.g., CIFAR 10 and Stanford dogs) used in this work are quite outdated. Numbers on its method is on par with fully-retrained models.

**Methods And Evaluation Criteria:**

Proposed method is based on Newton updates and DP noise, which are two important components been used in machine unlearning area, so it makes sense.

**Other Comments Or Suggestions:**

Typos on Line 150 “athat”.

**Other Strengths And Weaknesses:**

N/A

**Questions For Authors:**

I am not sure that the difference between DP and DP-inspired unlearning, even after reading appendix B. Why is the statistical indistinguishability of two approaches different? Can you clarify this point more in the text?

**Relation To Broader Scientific Literature:**

1. This work discusses some popular works on certified unlearning with different approaches. For example, Guo et al., 2019, Sekhari et al., 2021 and Zhang et al., 2024 focus on single-step Newton updates, and Neel et al., 2021 and Chien et al., 2024 focus on projected gradient descent algorithms.
2. As the closest problem to the proposed problem, authors discuss zero-shot unlearning approaches. Chundawat et al., 2023 requires the forget set, while this work does not need. Similarly, Cha et al., 2023 needs to apply gradient ascent on the forget set. This work also mentions Foster et al., 2024 and Bonato et al., 2025. However, none of them formally provide theoretical guarantee for the problem.

**Theoretical Claims:**

The theoretical claims look good.  Starting from Assumption 4.1 for loss function, this work develops  Hessian estimation, model update (with Newton updates) and Gaussian noise injection. Finally, it defines the bound between true model from the retraining and approximated model, and (epsilon-delta)-certified unlearning.

---

> ### Author Rebuttal · Authors · 2025-04-01
>
> We thank the reviewer for their constructive comments. Please see our responses below.
>
> **About the datasets:** It is worth noting that prior works in certified unlearning have similarly employed datasets such as CIFAR‑10 and StanfordDogs [R5-R7]. Our primary objective in using these datasets was to provide a controlled experimental validation of our theoretical results rather than to claim state‑of‑the‑art empirical performance.The fact that our method performs comparably to fully‑retrained models confirms that our unlearning approach preserves model utility while offering certification guarantees.
>
> [R5] A. Golatkar, A. Achille, A. Ravichandran, M. Polito, and S. Soatto, ‘Mixed-Privacy Forgetting in Deep Networks’, in Proceedings of the IEEE/CVF Conference on Computer Vision and Pattern Recognition (CVPR), 2021, pp. 792–801.
>
> [R6] E. Chien, H. P. Wang, Z. Chen, and P. Li, ‘Langevin Unlearning: A New Perspective of Noisy Gradient Descent for Machine Unlearning’, in The Thirty-eighth Annual Conference on Neural Information Processing Systems, 2024.
>
> [R7] B. Zhang, Y. Dong, T. Wang, and J. Li, ‘Towards Certified Unlearning for Deep Neural Networks’, in Forty-first International Conference on Machine Learning, 2024.
>
> **Difference between DP and DP-inspired unlearning:** The key difference between DP and DP‑inspired unlearning lies in the nature of their statistical guarantees. In traditional Differential Privacy (DP), the guarantee is based on the statistical distance between the output distributions of an algorithm when run on two adjacent datasets. In contrast, DP‑inspired unlearning focuses on ensuring statistical indistinguishability between the output of a fully‑retrained model and the output of the unlearning mechanism. This distinction means that while both approaches aim to limit the leakage of sensitive information, DP‑inspired unlearning calibrates noise based on the distance between these two different mechanisms rather than on adjacent datasets. We elaborate on this point further in Appendix B.
>
> **Typo:** Thanks for pointing this, we will fix this in the revised version.

---

### Official Review · Reviewer_mUHz · 2025-03-11

**Overall Recommendation:** 2

**Summary:**

In this paper, the authors studied an unlearning problem with the assumption that at the time of unlearning, the model provider lost access to the original training dataset, but has access to a surrogate dataset that’s very close to the original training dataset in terms of distribution. They adapted the second-order Newton update algorithm to perform unlearning and also provide theoretical guarantees for their proposed method. Moreover, they conducted preliminary experiments to demonstrate the performance of their proposed method on synthetic generated datasets and real-world datasets.

**Claims And Evidence:**

See my main comments in **Theoretical Claims** and **Experimental Designs Or Analyses** sections.

**Essential References Not Discussed:**

NA

**Experimental Designs Or Analyses:**

1. In the experimental results presented in Tab 1, it seems like the model after unlearning performs similarly at test and and forget data splits. In this case, does that means the test data has a distribution shift? Or the original model is not sufficiently trained on the “desired to learn” distribution? Moreover, it would be good to provide the original model performances on those data split, so we have a good understanding of the unlearn algorithms’ performance and also if unlearning makes sense in this setting.
2.	Since there is randomness introduced by noise calibration, it is good to repeat the experiments multiple times and report the means and standard deviations for all experiments.
3.  Instead of unlearning a randomly sampled subset of the original dataset, is it possible to unlearn a certain class from the original dataset, or unlearn all poisoned samples from the original dataset? In these problem settings, it seems to be easier to evaluate the performance of the proposed algorithm.

**Methods And Evaluation Criteria:**

1. Most of the surrogate datasets in the experiments are synthetically generated, does it mean that for real-world usecases, surrogate datasets should also be generated in this way? Can we use the train model to understand the original distribution and generate surrogate dataset based on model prediction?

2. Can the authors provide a few usecase examples when one would have accessed or created a surrogate dataset and lost access to the original dataset in real-world scenarios? I think the motivation for this problem setting is not clear.

**Other Comments Or Suggestions:**

1.	Page 3 line 150, "åthat" should be  "that"?

**Other Strengths And Weaknesses:**

Na

**Questions For Authors:**

See my main comments in **Theoretical Claims** and **Experimental Designs Or Analyses** sections.

**Relation To Broader Scientific Literature:**

The paper studied unlearning in a specific problem scenario, however, I am not very convinced that the problem setup is realistic, see my comments in **Methods And Evaluation Criteria** section.

**Theoretical Claims:**

I have some questions about the assumptions stated in this paper.

1.	Assumption 4.1 though may be typical for the second-order type of methods, it is very strong assumption on the loss function. To me, I think common loss functions used for training deep neural networks do not satisfy assumption 4.1, which makes the theoretical result stated in this paper unrealistic.
2.	I am a bit confused about the assumption of not having access to the original training dataset. What I understand is one does not have access to the full original dataset $\mathcal{D}$, however, it seems like the authors only assume that one does not have access to the retain portion of the original dataset $\mathcal{D_r}$,
which seems to make the problem easier, at least computing an approximation of $H_{\mathcal{D}_r}$ is easier under assumption 4.1 and with the knowledge of $\mathcal{D}_u$.

---

> ### Author Rebuttal · Authors · 2025-04-01
>
> We thank the reviewer for their constructive comments.
>
> **Justification of the motivation:** In our experiments, most surrogate datasets were synthetically generated to provide a controlled environment for evaluating our theoretical results. However, for real-world applications, surrogate datasets need not be generated purely synthetically. In practice, surrogate data can be obtained from publicly available sources or even generated by leveraging the trained model to capture aspects of the original distribution.
>
> Regarding the motivation, there exist numerous real-world scenarios where the original dataset is no longer accessible. For example, regulatory policies may require data deletion after a fixed retention period, while models trained on this data continue to be deployed. Similarly, in collaborative or federated learning settings, data privacy constraints often prevent access to the full original dataset, leaving only a surrogate or limited statistical summary available.
>
> Finally, we clarify the assumption regarding access to the original training dataset. In our framework, we assume that the unlearning mechanism does not have direct access to the retain portion of the original dataset ($\mathcal{D}_r$). It only has access to the samples to be forgotten ($\mathcal{D}_u$), which may for instance be provided by a specific user. Although one might expect that having access to the forget samples ($\mathcal{D}_u$) would simplify approximating the Hessian for the retained data ($\mathcal{D}_r$), in practice, this is nontrivial. The distribution of $\mathcal{D}_u$ can differ significantly from that of $\mathcal{D}_r$ (as in class unlearning), making Hessian estimation challenging when relying solely on the forget samples. This reflects our intent to keep the unlearning framework broadly applicable.
>
> **Assumptions on loss functions and practicality:** Yes, we agree with the reviewer that Assumption 4.1 is indeed strong; while it may not hold for all deep neural network loss functions, it is important for tractability of the theoretical analyses of second-order unlearning methods which is a key first step towards the analysis of the more general setting. To satisfy these conditions while still achieving competitive performance in practice, we leverage a mixed linear network [R4] implementation. Mixed linear networks combine the simplicity and theoretical tractability of linear models with select non-linear components, which enables them to meet the convexity and smoothness criteria. As demonstrated in the table below, our implementation achieves a significant unlearning performance depending on the accuracy metrics while also achieving significant training accuracy. Extending our theoretical guarantees to general non-convex settings in the source-free setting is an important future direction.
>
> [R4] A. Golatkar, A. Achille, A. Ravichandran, M. Polito, and S. Soatto, ‘Mixed-Privacy Forgetting in Deep Networks’, in Proceedings of the IEEE/CVF Conference on Computer Vision and Pattern Recognition (CVPR), 2021, pp. 792–801.
>
> ||Method|Train|Test|Retain|Forget|MIA|RT|$\Delta$|$\hat{\Delta}$|
> |-|-|-|-|-|-|-|-|-|-|
> ||Original|94.7±0.2%|88.5±0.3%|95.3±0.2%|95.0±0.4%|||||
> |0.1|Retrain|93.6±0.4%|86.4±0.9%|95.6±0.7%|84.7±0.6%|51.2±0.1%|53±2|||
> ||Unlearn(+)|93.7±0.3%|86.4±0.6%|94.8±0.4%|87.2±0.6%|51.3±0.1%|54±3|0.02||
> ||Unlearn(-)|94.1±0.3%|85.2±0.3%|94.9±0.7%|86.8±0.1%|52.1±0.8%|54±2|0.31±0.06|0.35±0.03|
> |0|Retrain|81.7±0.4%|72.3±0.1%|92.7±0.1%|0%||142±3|||
> ||Unlearn(+)|82.2±0.4%|72.5±0.9%|93.2±0.2%|4.2±0.2%||135±5|0.02||
> ||Unlearn(-)|82.4±0.5%|72.4±0.1%|93.5±0.8%|5.1±0.8%||132±6|0.49±0.03|0.52±0.02|
>
> **Class unlearning:** Thanks for raising this interesting question. Yes, it is possible to unlearn an entire class. We conducted experiments to unlearn class 0 from the CIFAR10 dataset, using our mixed linear network implementation, which satisfies the convexity assumptions while maintaining competitive accuracy. We show our results in the second row of the table above. We observe that we can effectively remove the influence of the targeted class while preserving performance on the remaining data. We believe the proposed method not only facilitates class unlearning but also can be extended to tackle scenarios such as unlearning all poisoned samples.
>
> **Original model results:** We added the original model results in every table we provided in this rebuttal (tables under Reviewer a1HE and the table provided above).
>
> **Experiments with different seeds:** To address the inherent randomness introduced by noise calibration, we conducted our experiments over multiple runs and report the averaged values with their error margins (table under Reviewer a1HE, the table provided above, Figure 1-4 and Table 1-2 in [link](https://limewire.com/d/a68qm#RoLgJHE3KN)). This evaluation demonstrates that our method is robust with performance variations well within acceptable margins.
>
> **Typo:** Thanks, we will fix this in the revised version.

---

### Official Review · Reviewer_a1HE · 2025-03-11

**Overall Recommendation:** 4

**Summary:**

This paper studies certified unlearning in a setting where the original training data is inaccessible. Prior work on certified unlearning guarantees that the unlearned model is statistically indistinguishable from a model retrained without the deleted data, however it requires access to the original training data to perform the update. To get around this limitation, the authors of this paper assume access to a surrogate dataset and provide bounds on indistinguishablility based on the total variation (TV) distance between the original training data distribution and surrogate data distribution. This TV distance is also used to calibrate the noise added to the model parameters to ensure indistinguishability. Since the TV distance cannot be computed in practice (as it depends on the unknown training data distribution), the authors propose a heuristic method to estimate it using the original model and a model trained on surrogate data. The approach is evaluated empirically on real and synthetic datasets, where it is found to maintain utility and privacy guarantees.

**Claims And Evidence:**

- The paper claims that achieving certified unlearning without training data access is an open problem. I agree based on my knowledge of the literature.

- The paper claims that the proposed method is certified, however this is only true for the method described in Sec 4.1 which is not realizable in practice. While the practical approach described in Sec 4.2 appears to perform well empirically, the bound on the unlearning error $\hat{\Delta}$ is not guaranteed to hold.

- The paper claims that extensive experiments demonstrate the effectiveness of the approach, however I have some concerns (see below).

**Essential References Not Discussed:**

I’m not aware of any essential references that were not discussed.

**Experimental Designs Or Analyses:**

As stated above (Methods and Evaluation Criteria) the experiments broadly make sense to me. However, I have the following concerns:



- I couldn’t spot the size of the forget set in Sec 5. I suspect this may have an impact on the performance of Unlearn $-$. It would be great to see results reported as a function of the size of the forget set.

- The error in the estimate of the KL divergence between the training and surrogate data does not seem to be reported. If this information were available, it would be possible to more directly assess the validity of the estimation approach described in Sec 4.2.

- The empirical estimate of the unlearning error $\hat{\Delta}$ (Corollary 4.4) is not reported. As a result, it’s not possible to assess whether the “approximate” certificate is practically useful, or vacuous.

- Fig 1(a): I believe the curve corresponds to the exact estimate of the required variance (assuming training data access)? If so, it would be good to additionally report the heuristic estimate (not assuming training data access).

- For the synthetic experiments, it would be good to repeat the experiment multiple times averaging over multiple synthetic datasets. The results could be reported with error bars.

**Methods And Evaluation Criteria:**

- The proposed method builds on Sekhari et al. (2021), except that the original training dataset is replaced with a surrogate. This makes sense, provided a close surrogate is available (I’m not sure how reasonable this assumption is). It makes sense to me to account for the error in approximating the training dataset with the surrogate, however, I wonder whether it would be possible to estimate an upper bound on the TV distance, rather than using an estimator with unknown properties. If this were possible, the “practical” method could be certified.

-  The evaluation broadly makes sense. The proposed “practical” method (Unlearn $-$) is compared with two baselines with training data access: Sekhari et al. (2021) and retraining from scratch. A variety of standard metrics for assessing unlearning performance are used across four datasets (three real, one synthetic). Due to strong assumptions on the loss function (Assumption 4.1), the evaluation focuses on linear models or pre-trained encoders with a learned linear head. This is also a limitation in prior work.

**Other Comments Or Suggestions:**

N/A

**Other Strengths And Weaknesses:**

S1. I found the paper a pleasure to read.

S2. The problem seems well-motivated given the existence of several heuristic approaches that aim to perform unlearning without training data access.

S3. Apart from the suggestions above, the experiments seem well-executed.

W1. The paper focuses on single batch unlearning for classification models, which limits its applicability. However, this also appears to be a limitation of prior work.

**Questions For Authors:**

1. How does the method perform for different sized forget sets?

2. How large is the empirical bound on the unlearning error? Is it practically useful?

3. Is it possible to approximate a (non-vacuous) upper bound on the KL divergence?

**Relation To Broader Scientific Literature:**

- The paper builds on Sekhari et al. (2021), adopting their definition of certified unlearning and adapting their unlearning method. Sekhari et al.’s formulation of unlearning differs from prior work by Guo et al. (2019) and Neel et al. (2021), in that they focus on ensuring the unlearned model achieves a high (generalization) test error rather than training error. Heuristic approaches for unlearning without access to training set.

- The practical version of the method estimates the KL divergence from the original model and a model trained on surrogate data using:

    - An implicit energy model to estimate the marginal distribution over inputs $P(X)$ (Grathwohl et al. (2019).

    - A Donsker-Varadhan variational representation of the KL divergence (Donsker & Varadhan, 1983).

- The empirical evaluation employs a variety of unlearning metrics by Kurmanki et al. (2023), Golatkar et al., (2020) and Triantafillou et al., (2024).

**Theoretical Claims:**

The main theoretical claim is stated in Theorem 4.2 and proved in Appendix C. I did not check the proof, however it seems to make use of the triangle inequality, various assumptions on the loss function, and properties of the Hessian. I don’t have any reason to doubt the correctness.

---

> ### Author Rebuttal · Authors · 2025-04-01
>
> We thank the reviewer for their constructive comments.
>
> **Experiments with different forget ratios:** We conducted extensive experiments on the StanfordDogs dataset with varying forget ratios to assess how forget set ratio impacts unlearning. The results in the table below show that our method Unlearn(-) scales well across different forget set ratios (FR). Also, results under the MIA and RT columns indicate that similar unlearning performance is achieved across FR with Unlearn(+) (the model achieved after unlearning with the exact data samples) and Retrain models. These findings confirm the robustness of our approach.
>
> |FR|Method|Train(%)|Test(%)|Retain(%)|Forget(%)|MIA(%)|RT|$\boldsymbol{\Delta}$|$\boldsymbol{\hat{\Delta}}$|
> |-|-|-|-|-|-|-|-|-|-|
> |0.01|Original|87.3±0.2|73.7±0.3|87.2±0.2|87.9±0.4|||||
> ||Retrain|87.1±0.1|73.7±0.7|87.2±0.3|73.8±0.6|52.1±0.2|10±2|||
> ||Unlearn(+)|87.3±0.4|74.1±0.6|87.3±0.1|74.5±0.7|53.2±0.3|10±2|0.0002||
> ||Unlearn(-)|87.1±0.6|74.1±0.4|87.2±0.3|74.1±0.6|53.1±0.2|10±1|0.023±0.006|0.032±0.007|
> |0.2|Original|87.3±0.2|73.7±0.3|87.6±0.2|86.4±0.4|||||
> ||Retrain|85.6±0.1|72.4±0.6|88.7±0.7|73.3±0.1|50.6±0.8|40±2|||
> ||Unlearn(+)|84.9±0.8|71.8±0.5|88.3±0.1|71.5±0.3|51.8±0.2|40±1|0.08||
> ||Unlearn(-)|85.0±0.5|71.4±0.2|88.0±0.9|72.6±0.6|52.0±0.1|40±2|0.82±0.06|0.91±0.03|
>
> **Justification of the heuristic method and practicality:** We conducted experiments on both synthetic and real datasets to justify the heuristic KL‐divergence estimation (Sec. 4.2) and the corresponding empirical unlearning error $\hat{\Delta}$. In Figs. 1 and 3 [(link)](https://limewire.com/d/a68qm#RoLgJHE3KN), we plot both the “exact” and “approximated” results (our heuristic method) for the KL‐divergence and the respective noise $\sigma$. For the synthetic data experiments, the exact KL divergence is computed using its closed-form for Gaussians. For real data experiments, since the exact KL divergence was not available, we estimated it using the Donsker-Varadhan bound as a reference, leveraging both exact and surrogate data samples. Figs. 1–4 and Tables 1–2 [(link)](https://limewire.com/d/a68qm#RoLgJHE3KN) show our approximations closely match exact values.
>
> To address whether the approximate certificate is practically useful, we report both $\Delta$ (exact) and $\hat{\Delta}$. The results confirm that even if the estimated bounds grow, model performance aligns with Unlearn(+). For completeness, we ran multiple trials with different seeds, included error bars in all figures, and included error margins into the tables demonstrating consistency across repeated experiments. Overall, these findings validate the heuristic’s reliability and practical utility in estimating KL‐divergence and unlearning error.
>
> **Non-vacuous upper bound for statistical distances:** While a trivial upper bound on the TV distance is available ($TV(\rho \| \nu) \leq 2$), extending this to a non-vacuous bound on the KL divergence is challenging without further assumptions on the retained data. Our primary goal was to maintain generality and avoid introducing new restrictions with the assumptions about the data distributions. To that end, we proposed a heuristic method to approximate the KL distance using the model parameters. Using this approximation, showed that as the distance between exact and surrogate distributions increases, achieving certified unlearning requires adding more noise calibrated using Corollary 4.4. Figs. 1 and 3 [(link)](https://limewire.com/d/a68qm#RoLgJHE3KN) further show that our method can approximate the exact KL distance using only the model parameters, without access to the original data samples, making it suitable for source-free unlearning.
>
> **Limitations:** Regarding the assumptions, we build on the mixed linear network which enables us to meet the strong convexity criteria while also achieving competitive accuracy results. We believe that this choice satisfies both the necessary theoretical conditions and provides a solid foundation for our experimental evaluation. Please also see our response to Reviewer mUHz, where we provide further details about this network under "Assumptions on loss functions and practicality".
>
> While our current formulation focuses on single-batch unlearning for classification models as noted by the reviewer, this also provides an important and tractable first step. Extending our mechanisms to sequential unlearning requires new theoretical insights on the certification budget considering the noise added after each unlearning step [R2, R3]. This is an interesting future direction we are actively exploring.
>
> [R2] E. Chien, H. P. Wang, Z. Chen, and P. Li, ‘Langevin Unlearning: A New Perspective of Noisy Gradient Descent for Machine Unlearning’, in The Thirty-eighth Annual Conference on Neural Information Processing Systems, 2024.
>
> [R3] B. Zhang, Y. Dong, T. Wang, and J. Li, ‘Towards Certified Unlearning for Deep Neural Networks’, in Forty-first International Conference on Machine Learning, 2024.

---

> > ### Comment · Reviewer_a1HE · 2025-04-02
> >
> > Thanks for addressing my questions around the experimental evaluation. I'm satisfied with the response and will update my score.

---

> > > ### Author Response · Authors · 2025-04-02
> > >
> > > We are glad to have answered your questions, thanks for the insightful comments.

---

### Official Review · Reviewer_jhwC · 2025-03-13

**Overall Recommendation:** 3

**Summary:**

This paper proposes a novel certified unlearning framework that enables the unlearning process without requiring access to the original dataset. The authors use an estimated surrogate dataset Hessian to approximate the second-order unlearning process. The surrogate dataset is generated by estimating the model's original target distribution and using SGLD and Donsker-Varadhan techniques to sample and estimate the distribution of the new surrogate dataset. The work also provides a theoretical upper bound proof that supports the theoretical correctness of the proposed method.

**Claims And Evidence:**

yes

**Essential References Not Discussed:**

No

**Experimental Designs Or Analyses:**

Sufficient validations on various datasets and model architures. Though the test architectures and dataset complexity are very simple.

**Methods And Evaluation Criteria:**

Yes, the authors provides results on various real world datasets and synthetic datasets

**Other Comments Or Suggestions:**

No

**Other Strengths And Weaknesses:**

Pros:
1. The paper addresses an important issue by introducing a framework for unlearning without access to the original dataset.
2. The use of surrogate datasets and second-order unlearning seem novel
3. The theoretical analysis provides a strong foundation for the proposed method.


Cons:
1. The sampling and estimating the surrogate dataset part could be presented earlier.
2. The network's architecture used in the empirical is too simple. I wonder the effectiveness when apply in models with moderate complexity.

**Questions For Authors:**

How did you compute the noise variance and forget scores for various datasets shown in Figure 2?

**Relation To Broader Scientific Literature:**

The use of surrogate datasets and second-order unlearning is interesting and promising, can be beneficial to broader context.

**Theoretical Claims:**

Appear correct

---

> ### Author Rebuttal · Authors · 2025-04-01
>
> We thank the reviewer for their constructive comments. Please see our responses below.
>
> **Experiments on complex models:** Our main focus in the current work was the theoretical foundations, to explore provable certified unlearning mechanisms in the source-free setting, and provide rigorous theoretical guarantees for certified unlearning using surrogate datasets. To that end, our empirical evaluations ranged - as a proof-of-concept - from simple linear models to networks with one or two convolutional layers (see Table 5 in the paper) that demonstrate that our certified unlearning mechanism maintains competitive performance across various complexity levels.
>
> To further address the concern about moderately complex architectures, we introduce in this rebuttal a mixed linear network implementation. This model satisfies the strong convexity  assumptions made in our analysis—particularly regarding the loss function—and provides a more expressive and realistic evaluation setting than standard linear models. Despite its increased complexity, it remains analytically tractable and directly reflects the practical applicability of our theory. We gave more details about mixed linear networs in our responses to Reviewer mUHz under “Assumptions on loss functions and practicality”. As shown in the table given under the same response, our certified unlearning mechanism performs effectively in this mixed linear setting. This strengthens our claim that the theoretical results extend meaningfully to models beyond simplistic settings, including those with moderate complexity.
>
> In response to the reviewer’s comment, we evaluate our method on a ResNet-18 model by computing the full-model Hessian using the Conjugate Gradient method. The results are shown in the table below. We observe a performance drop on the forget set after the unlearning update for both Unlearn (+) (retrained with exact data) and Unlearn (−) (our method). Importantly, Unlearn (−) performs comparably to Unlearn (+), demonstrating that our method achieves similar unlearning behavior without access to the retain dataset, even on complex models like ResNet-18.
>
> |Method|Train|Test|Retain|Forget|MIA|RT|
> |-|-|-|-|-|-|-|
> |Unlearn(+)|96.5%|90.4%|97.1%|94.4%|56.2%|45|
> |Unlearn(-)|96.1%|89.7%|96.2%|94.2%|57.3%|49|
>
> **Noise variance and forget score:**
>
> - Noise Variance ($\sigma$): Our method derives $\sigma$ based on the  theoretical bounds provided in Theorems 4.2 and 4.3 using an upper bound on the statistical distance between the true data distribution and the surrogate distribution, which control the term $\|w_{r}^* - \hat{w}_r\|_2$ (see Equation (8)). In practice, when we do not know the exact upper bound on the statistical distance between the two datasets, we estimate it using the heuristic method explained in Sec. 4, and this estimate is then substituted into Equation (6) to compute the calibrated noise variance. To do so, we first sample from the trained model parameters using Langevin dynamics (details in App. D3) and then estimate the KL distance using the Donsker-Varadhan bound (Equation (12)).
> - Forget Scores: To quantify the quality of forgetting, we compute per-example $\epsilon$ estimates that capture the strength of an optimal attacker's ability to distinguish between the outputs of the unlearned model and a model retrained from scratch. These per-example estimated $\epsilon$ values are then aggregated into an overall forget score using a binning approach to ensure granularity and robustness. Our forget score methodology and implementation follows along the lines of Triantafillou et al. [R1], and we report three variants: “FS1(+)” (using noise calibrated with the original data), "FS1(-)” (using noise calibrated with the surrogate data, our method), and “FS2(-)” (adding noise calibrated with the original data but applied without re-calibration to the surrogate data).
>
> We hope this clarifies how the noise variance and forget scores are computed for the datasets in Figure 2 in our paper.
>
> Finally, we agree with the reviewer’s suggestion on moving the surrogate dataset part earlier, we will present this part earlier in the revised version.
>
> [R1] Triantafillou, E., Kairouz, P., Pedregosa, F., Hayes, J., Kurmanji, M., Zhao, K., Dumoulin, V., Junior, J. J., Mitliagkas, I., Wan, J., et al. Are we making progress in unlearning? Findings from the first NeurIPS unlearning competition,  arXiv:2406.09073, 2024.

---

### Decision · Program_Chairs · 2025-05-01

**Decision:**

Accept (poster)

**Comment:**

This paper consider machine unlearning, an important task motivated by the right to erasure present in a number of jurisdictions. While the motivation is connected to privacy and data minimisation, the predominant approach to machine unlearning assumes access to the entire training set, in order to correct for accumulating noise required for indistinguishability. That is, the main benefit delivered by most existing work is computational, and in offering an approach to unlearning, actually requires *greater* access to training data. Given that many companies tend towards data minimisation, this is a significant gap in the practicality of machine unlearning – computational advantage at the expense of privacy. This work develops interesting approaches that don’t require access to training data but instead leverage surrogate data. Noise is then calibrated according to the statistical (TV) distance between training and surrogate data, allowing for rights to privacy and erasure to coexist.

In response to concerns by Reviewer jhwC on simplicity of models evaluated, the authors present convincing experimental evidence in their rebuttal on a ResNet-18 model, showing strong comparable performance against the natural baseline. The other concerns such as estimating of variance by an upper bound on statistical distance between true and surrogate distributions, and forget scores, are fully addressed by the authors highlighting existing explanation in the paper.

Reviewer mUHz reports a range of concerns spanning the motivation, with most of the critique clarifying in nature: whether surrogate data needs to be synthetic; Assumption 4.1 vs “typical” loss function; clarification of components of training detail retained; interpretation of Table 1. And relatively minor points: repeats to account for randomness in experiments, unlearning classes/other settings.

Reviewer a1HE raises several valid areas for improvement, much of which are addressed by the author rebuttal: size of forget set in Sec 5; error analysis of KL estimate; empirical estimate of unlearning error $\hat{\Delta}$; heuristic estimate vs exact estimate of the required variance; averaging for synthetic experiments to obtain error bars. The authors must implement these improvements and clarifications, I think them for the responses so far.

A key concern highlighted by a1HE is that the unlearning approach, though certified for exact TV distance, loses its guarantees of certified unlearning in practice. This is a fairly important distinction that needs to be highlighted better in the abstract and introduction, to make clear the potential disconnect. That said, the Reviewer ultimately found the work valuable – as indeed much of the literature between practical unlearning and certified unlearning is still addressing this gap, and I believe the work does represent progress towards this goal, as the requirement to retain training data can be seem to undermine the privacy rights ultimately motivating or connected to the right to erasure.